# Mechanically Induced Pulpitis: A Rat Model That Preserves Animal Well-Being

**DOI:** 10.3390/biomedicines13081925

**Published:** 2025-08-07

**Authors:** María Alexandra Bedoya, Gloria Cristina Moreno, Camilo Durán, Adriana Camacho, Angel Eduardo Pirela, Stefany Rojas Lozano, Maddy Mejía, Eddy Herrera, Luz-Stella Rodríguez Camacho, Lorenza Jaramillo, Nelly S. Roa

**Affiliations:** 1Centro de Investigaciones Odontológicas, Faculty of Dentistry, Pontificia Universidad Javeriana, Bogotá 110231, Colombia; bedoya-a@javeriana.edu.co (M.A.B.); gcmoreno@javeriana.edu.co (G.C.M.); camilo.duran@javeriana.edu.co (C.D.); camacho-adriana@javeriana.edu.co (A.C.); apirela@javeriana.edu.co (A.E.P.); stefanyrojas1717@gmail.com (S.R.L.); 2Departamento de Patología, Hospital Universitario San Ignacio, Pontificia Universidad Javeriana, Bogotá 110231, Colombia; maddy.mejia@javeriana.edu.co; 3Física Matemática, Departamento de Matemáticas, Facultad Ciencias, Pontificia Universidad Javeriana, Bogotá 110231, Colombia; eherrera@javeriana.edu.co; 4Instituto de Genética Humana, Facultad de Medicina, Pontificia Universidad Javeriana, Bogotá 110231, Colombia; luz-rodriguez@javeriana.edu.co

**Keywords:** mechanical pulp exposure, mechanically induced pulpitis, dental pulp, animal well-being, rat model, animal model, pulpitis

## Abstract

**Background**: Understanding the mechanisms underlying dental pain caused by pulpitis in humans has led to the development of animal models, such as the rat, which enable the study of the mechanisms underlying inflammation; the use of these models is considered ethically justified when the anticipated scientific benefits outweigh the potential impacts on animals in the harm/benefit balance. **Objective**: To develop a rat model of mechanically induced pulpitis and to evaluate the potential impact on animal well-being. **Methods:** Pulpitis was mechanically induced in male Lewis rats (13–16 weeks, 350–400 g) which were anesthetized and endotracheally intubated. Following pulp exposure, the cavity was sealed with either amalgam (n = 10) or zinc phosphate cement (n = 10). Following recovery and return to their housing, behavioral assessments and histological evaluations using Hematoxylin and Eosin (H&E) staining were conducted in separate cohorts at two time points: 3 h and 5 days following the procedure. **Results**: A standardized model of mechanically induced pulpitis was established and verified clinically and by histopathological analysis, which showed evidence of the inflammatory process and revealed no statistically significant differences in the scoring of pain, discomfort, or distress, nor in the measurements of food and water consumption or body weight. **Conclusions**: The behavioral assessments conducted in this study supported the implementation of a safe and easily reproducible model for future research aimed at elucidating the mechanisms underlying pulp inflammation.

## 1. Introduction

Dental pulp is a loose connective tissue rich in blood vessels and nerves that ensure tooth vitality [1]. This highly specialized tissue contains a network of CD45+ immunocompetent cells, responsible for maintaining tissue homeostasis and immunoregulation [2], as well as myelinated and unmyelinated polymodal Aδ and C nerve fibers. In response to noxious stimuli, these fibers synthesize and release neuropeptides such as substance P, calcitonin gene-related peptide (CGRP), Neurokinin A (NKA), Neuropeptide Y (NPY), and Vasoactive Intestinal Peptide (VIP) [3]. The release of these neuropeptides and their interactions with specific receptors mediate and regulate blood flow, neurogenic inflammation, and the transmission of painful stimuli, contributing to the progression of pulp lesions and the development of dental pain [3,4].

When injured, whether by caries, trauma, excessive wear, or dental procedure, the pulp initiates an inflammatory response known as pulpitis [5]. As in other tissues, inflammation is a prerequisite for proper healing [6], involving sequential phases homeostasis, inflammation, proliferation, and tissue remodeling [7]. However, to maintain functional teeth throughout life, vital and healthy dental pulp is required [8,9,10].

Pulp removal as a treatment [8], has encouraged interest in the development strategies for the revitalization and regeneration of the pulp. Research in pulp therapy increasingly focuses on modulating the biological mechanisms of inflammation. A wide range of experimental models are used, with in vivo models offering the advantages of clinically assessing the actual inflammatory state of the pulp [5]. The diversity of available animal models highlights the challenge of selecting the most appropriate model [5,11]. The rat molar has been proposed as a suitable model for studying pulpitis due to its anatomical, histological, and physiological similarities to the human molar [12,13]. The essential biological responses and stages of pulp healing in rats are comparable to those in other mammals [14].

In these models, inflammation is typically induced by perforating the occlusal surface of the molar to expose the pulp to the oral environment [5]. In some cases, the cavity is immediately sealed with dental materials [15,16,17] while in others, it is left open to allow for microbial exposure identified in previous reports as open cavity models [18,19,20]. Open cavity models generally induce progressive inflammation due to direct exposure of the pulp to oral microorganisms, exacerbating inflammation and activating nociceptive pathways associated with pain-related behavioral changes [20,21]. Sealed cavities models aim to protect the pulp from microbial invasion and provide a controlled environment for testing therapeutic agent. However, the chemical nature of the sealing material can modulate the intensity of the inflammatory response and affect experimental outcomes, making it essential to preserve animal well-being throughout the study.

Some studies have reported that sealed cavity models do not significantly alter parameters such as body weight, food and water consumption, general behavior, or systemic inflammation [12,20]. However, few studies have systemically evaluated the impact of these models on animal well-being. In some cases, local inflammatory agents such as Complete Freund’s Adjuvant (CFA), have been used to induce local inflammation, which has been shown to impair animal locomotion for up to eight days following the injury [20].

Given the ethical imperative to evaluate the benefits of animal research against potential well-being impacts and in accordance with the 3Rs principle: Replacement, Reduction and Refinement [22,23], it is crucial to implement strategies that minimize pain and distress. This requires rigorous experimental planning and review to ensure that each study contributes substantially to scientific knowledge and ultimately benefits both human and animal life by supporting the goal of maintaining or improving their well-being [23]. The objective of this study was to develop a model of mechanically induced pulpitis in the lower right first molar of *Lewis* rats, using either amalgam or zinc phosphate as sealing materials and to assess the impact on animal well-being. It is hypothesized that, due to their chemical nature, amalgam or zinc phosphate as sealing materials may induce different types and prolonged inflammatory responses, in addition to the induced pulpitis, with amalgam causing mild and chronic inflammation [24], and zinc phosphate leading to severe or advanced inflammation [25,26,27]. Based on this, a physical and chemical pulp injury free of pathogens was induced.

Behavioral changes were evaluated at two time points, at 3 h and 5 days post-induction using a well-being scoring system, along with measurements of body weight and food and water consumption.

## 2. Materials and Methods

### 2.1. Animals

This study was approved by the Institutional Animal Care and Use Committee (IACUC) of the Pontificia Universidad Javeriana (FUA107-21), which is part of a program fully accredited by AAALAC International, and by the Research and Ethics Committee of the Faculty of Dentistry (CIEFOPUJ) (OD-216). Prior to the main study, a pilot study was conducted and approved (C-222-18). A total of 54 male Lewis rats (LEW/Crl), aged 13–16 weeks and weighing 350–400 g, were used. The animals were housed under Specific pathogen-free (SPF) conditions in autoclavable Zyfone microisolators cages (Lab Products, ONE CAGE 2000). Environmental conditions were maintained at 22 ± 3 °C, 50 ± 20% relative humidity, with a 12 h light/dark cycle and 50 filtered air exchanges per hour (Trane, Minisplit), in a noise and odor-controlled room. The microenvironment included absorbent bedding (Souralit 7/4S, Aspen Biotech, España) and weekly changed enrichment materials (Souralit C32/23, Aspen Biotech, España). Permanent enrichment (LifeSpan™ Rodent Enrichment, Lab Products, Y Corporation of America, Inc, Miami, USA) and alternating items such as chew sticks (Souralit Aspen Brick, Aspen Biotech, Barcelona, España) or tunnels (Souralit R75/3, España) were also provided. The rats were housed in groups of three per cage and had *ad libitum* access to an autoclavable diet (1013, LabDiet, Richmond, IN, USA) and filtered, autoclaved drinking water. To reduce stress and facilitate handling, all animals were habituated to researcher interaction using a tickling technique, performed daily for one week prior to the procedures. The study was conducted and reported in accordance with the ARRIVE (Animal Research: Reporting of In Vivo Experiments) guidelines.

Group Distribution. In the pilot study (n = 10), the rats were randomly assigned to two groups: one group underwent mechanical induction of pulpitis (MIP) under anesthesia with lidocaine (n = 5), and the other group underwent MIP without lidocaine (n = 5). Lidocaine was administered systemically during the procedure at a dose of 0.67–1.3 mg/kg/h, and again one-hour post-procedure at 0.17 mg/kg over 3 min for maintenance and control of pain, which provides controlled analgesia in animal models [28,29]. Within each group, a negative control (NC) and two restorative materials were evaluated: amalgam (DPFTPT-007, Nu. Alloy New Stetic, Guarne-Antioquia) and zinc phosphate cement (2011DM-001640R1, Lee Smith Cement, Industrial Latorre Caicedo, Bogotá, Colombia), referred to as MIPA and MIPZ, respectively. Additionally, a compound with therapeutic potential Capsaicin (CP) [100 μM] [30,31], was applied to the exposed pulp using a pipettor with a sterile tip (0.5 μL) prior to sealing the cavity with each material, resulting in the subgroups MIPA + CP and MIPZ + CP. CP was prepared one day prior to the experiment under sterile conditions in a laminar flow cabinet as follows: starting with a stock solution of CP in ethanol at 327.400 mM, 3.05 μL was diluted to 10 mL with sterile-filtered PBS (pH 7.4) and stored at 4 °C until use.

Following the pilot, it was determined that a more detailed evaluation of animal well-being was needed for the MIPA and MIPZ groups at two time points. The sample size was calculated using the G*Power 3.1.9.7 program, considering a significance level of 0.05 and a large, expected effect size (d = 0.8, according to Cohen). The analysis indicated that a sample size of n = 10 rats per group would provide 81% of statistical power to detect significant differences in the variables of interest between treatments (MIPA vs. MIPZ), both for immediate 3 h and delayed 5 days responses. Based on prior evidence showing no behavioral response in animals not subjected to pulpitis, a negative control group (NC, n = 2 per time point) was included for comparison. All male Lewis rats (350–400 g) were first enrolled based on the predefined inclusion weight range to limit variability and within that range, animals were randomly assigned to treatment arms using a computer-generated random sequence, with assignment concealed until immediately before the procedure.

Each time point was evaluated in separate cohorts: the first at 3 h to assess immediate response, and the second at 5 days to assess delayed response. The total number of animals used in the main study was 44.

### 2.2. Experimental Procedures

#### 2.2.1. Endotracheal Intubation

Endotracheal intubation was performed under general anesthesia in a rodent operating room maintained at 24 ± 2 °C, 50 ± 10% relative humidity, and 15 filtered air exchanges per hour. Anesthetic induction was initiated in a chamber which oxygen was administered at 2 L/min for 2 min, followed by 3% isoflurane for 3–4 min. Following induction, the rat was transferred to the intubation platform and positioned in a dorsal–ventral orientation at a 30° angle on a heating pad to maintain body temperature. Anesthesia was maintained with 3% isoflurane delivered via a face mask. Throughout the procedure, vital signs including body temperature, respiratory rate (RR), heart rate (HR), and peripheral capillary oxygen saturation (SpO_2_) were continuously monitored using a PhysioSuite^®^ system (PS-04-1EA, Kent Scientific, Torrington, CT, USA). Once physiological parameters stabilized, endotracheal intubation was performed using an LED headlight with a 3.5× magnifying telescope (Dr. Kim Headlight, Closter, NJ, USA, Innovaderma, Iver, UK), which provided clear, shadow-free visualization of the vocal cords. A 5 cm 16G catheter was then inserted through the vocal cords and connected to an automatic ventilator (Rovent^®^, Kent Scientific), integrated with the PhysioSuite^®^ system for continuous monitoring and respiratory support.

#### 2.2.2. Mechanical Induction of Pulpitis (MIP)

Following intubation, the rat was transferred to a custom designed flat platform, designed and manufactured by the research group (C.D.). The animal was positioned in a ventro–dorsal orientation on a heating pad to maintain body temperature. The skull was immobilized (head fixation) using adjustable auricular metal auricular stylets, and the oral opening was maintained using orthodontic posts and bands: the upper incisors were secured to an upper post, and the lower incisors to a lower post attached to the mandible. This setup, combined with the use of an LED headlight, enabled clear visualization of the molars. Trained operators created a standardized cavity on the occlusal surface of the lower right first molar using a pediatric 1/4 round bur (310213, SS White, Lakewood, NJ, USA), progressively advancing through the dentin until the pulp chamber was reached. Pulp exposure was confirmed with a No. 15 endodontic file. All procedures were performed by the same calibrated endodontist to ensure consistency across specimens. The cavity was then dried with paper points and sealed with either amalgam or zinc phosphate cement. The MIP procedure was refined through a coordinated teamwork of operators, who performed specific tasks including molar disinfection with 0.12% chlorhexidine (Periogard^®^, Colgate, Cali, Colombia) using disposable Micro Applicators (Global Roll^TM^), retraction of the facial muscle retraction using a dental spatula, activation of the handpiece with medicinal grade N_2_ to avoid air contamination, continuous irrigation with sterile 0.9% NaCl solution, continuous suction, tongue retraction with an ejector, and placement of the restorative material using an amalgam carrier and a micro-condenser for endodontic surgery. The dentist-endodontist ensured that the sealing material was adequately condensed and sealed inside the cavity, aiming to preserve the occlusal table and maintain the characteristic morphology of the first molar.

Throughout the procedure vital signs were continuously monitored, and the absence of pedal, tail, and palpebral reflexes was verified and recorded. After sealing the cavity, the isoflurane flow was gradually reduced to 0%. Once the reflexes began to return, the rat was extubated and monitored for 30 min during recovery before being returned to its original cage until the designated evaluation time. At 3 min or 5 h post-procedure, euthanasia was performed under deep anesthesia with 3–5% isoflurane, followed by CO_2_ inhalation.

### 2.3. Well-Being Assessment

In the pilot study, an ethogram was developed prior to the MIP experiment. Researchers were trained by systematically recording the spontaneous behaviors of male rats in their housing cages. Following the MIP procedure, the well-being assessment was conducted over a seven-day period by analyzing video recordings taken continuously during the first 24 h and daily thereafter. Video analysis was performed by two independents, blinded to treatment allocation, trained, and calibrated observers using validated dental pain assessment scales, including those proposed by Rifai [32], the Rat Grimace Scale [33], and Frecknell’s criteria [34]. Discrepancies, which accounted for less than 5% of the observations, were resolved through joint review and consensus based on video recordings. No significant discrepancies were observed between evaluators, and a kappa coefficient greater than 0.80 was obtained for all behaviors assessed. During this phase, the IACUC required a thorough and independent review of the recordings by multiple observers in all cases where a score of zero was assigned. These results of the pilot study were reviewed and validated by our IACUC, which subsequently authorized the continuation of the main study, allowing for the evaluation of rat behavior within the cage using forms incorporating the aforementioned scales.

In addition to behavioral analysis, routine colony monitoring included daily checks of food and water consumption, body weight, and the application of humane endpoint criteria by the facility’s veterinary staff when necessary.

In the main study (MIPA and MIPZ groups), well-being was assessed at two time points, 3 h and 5 days post-procedure. Quantitative variables included body weight and consumption of food and water. Body and food weights were measured using a precision rodent scale (Entris 98648-019-82, Sartorius^®^), and water intake was measured using sterile test tubes.

Additionally, a standardized well-being assessment form adopted by IACUC and based on the FELASA guidelines [35,36], was used to evaluate signs of pain, distress, or discomfort. This included clinical examination of appearance: piloerection, passivity, ocular discharge, closed eyelids, poor coat condition or alopecia and respiratory distress or increased respiratory rate; behavior: response to handling, gait, and social interaction; posture: hunching (lordosis, kyphosis) and immobility.

Each cage was observed directly for 30 min. Observations began with an external inspection followed by removal of the cage from the rack and its lid to allow for closer examination. Each variable was scored as (0) zero (normal/absent) or 1 (altered), with alteration further categorized by severity (mild, moderate or substantial). For the 3 h evaluation, assessments were conducted at three time points, one day before MIP, immediately before the procedure, and just prior to euthanasia. For the 5-day evaluation, assessments were conducted on the day before MIP, the day of the procedure, and on days 1, 2 and 5 post-procedure.

After euthanasia, the jaws were dissected, digital radiographs were used to confirm the presence of the restorative material, and pulp injury was verified by histological analysis using Hematoxylin and Eosin (H&E) staining. The microscopic analysis of the inflammatory state was performed by a pathologist from the department of pathology of the Hospital Universitario San Ignacio (HUSI), under simple blind observation conditions, using a light optical microscope (Olympus Optical). The analysis was performed on two sections for each dental pulp sample, at two time points (3 h and 5 days), across the MIPA, MIPZ, and NC experimental groups. Two parameters were evaluated: (1) Percentage of the coronal area affected by inflammatory infiltrate: Score 1: absent; Score 2: 1–30%; Score 3: 31–60%; Score 4: 61–100% and (2) Percentage of necrosis: Score 1: absent; Score 2: 1–10%; Score 3: 11–20%; Score 4: >20%.

### 2.4. Statistical Analysis

The inflammatory status of the pulp following treatment with amalgam (MIPA), zinc phosphate (MIPZ), and in the control group (NC) using two parameters: the percentage of coronal area affected by inflammatory infiltrate and the percentage of necrosis. Data were collected at two time points (3 h and 5 days post-treatment). Descriptive statistics were applied; for each sample, the score distribution was recorded, and the frequency percentages for each treatment group were reported. Continuous variables, including body weight, food consumption, water consumption, and their respective changes, were analyzed independently at two time points: (3 h and 5 days post-treatment). Only between the MIPZ and MIPA groups (n = 10 each), the negative control (NC, n = 2) was excluded from all inferential tests and used solely for descriptive baseline verification under the 3Rs principle. For each variable and time point, Shapiro–Wilk tests first assessed normality within each treatment. Levene’s test then evaluated homogeneity of variances between the two groups. When both normality and homogeneity of variances were satisfied, a two-sample Student’s *t*-test was applied; if variances were unequal, Welch’s *t*-test was used; and if normality was violated in either group, the Wilcoxon rank-sum test was employed. Response curves for each well-being indicator were generated from mean values at day 5, and gain/loss indices plus rates of body-weight change were computed per treatment. All statistical analyses were performed using R software (version 4.4.1 of 2024), under the GNU General Public License (GPL).

## 3. Results

### 3.1. Oral Opening and Dentition Characteristics

It was determined in this study that the oral opening is 23 ± 2 mm for rats of the assessed age and weight (Figure 1A). The dentition characteristics were as follows: in the upper jaw, there are two central incisors, and six molars located posteriorly, behind the palatal rugae, separated from the incisors by a 6 mm diastema. In the lower jaw, there are also two central incisors, separated from the molars by a 7 mm diastema. Visualization and access to the molars required cheek retraction (Figure 1B,C), allowing for the insertion of a high-speed handpiece with a 1/4 pediatric bur into the mesial fossa of the lower first molar (Figure 1D). Dissection of the hemi-maxilla confirmed the dental formula in rats as follows: central incisors (1/1), absence of lateral incisors (0/0), canines (0/0) and premolars (0/0) followed by molars (3/3), resulting in four upper and four lower teeth per hemi-maxilla, for a total of 16 teeth (Figure 1E,F).

Molar morphology was consistent across individuals. In the upper jaw, the first molar had five roots, two fossae and six cusps; the second molar, had four roots, two fossae, and five cusps; and the third molar had three roots, two fossae, and four cusps.

In the lower jaw, the first molar exhibited three vestibular cusps (Figure 1G) and three lingual cusps (Figure 1H), three fossae (Figure 1I,K) and four roots (Figure 1J,M). The second molar had three roots (Figure 1J), two fossae (Figure 1I) and five cusps (Figure 1H), while the third molar had two roots (Figure 1J), one fossa (Figure 1I) and four cusps (Figure 1H).

The pilot study results showed that, based on video analysis over a seven-day observation period (Figure 1Q–S), no behavioral indicators of compromised well-being were detected. The application of the Rifai, Grimace, and Flecknell scales yielded scores of zero, indicating no changes in facial expression or vibrissae orientation. Body weight evaluation showed no decrease; on the contrary, it increased steadily over time. Treatments with systemic lidocaine and the local capsaicin application did not result in observable behavioral changes in the evaluated rats. Based on these results, the IACUC approved proceeding without their use of the main study.

### 3.2. Implementation of the Mechanical Induction Model of Pulpitis

The mechanical induction of pulpitis (MIP) experiments conducted in this study enabled the implementation of a model that avoids the use of analgesics, which can often interfere with experimental outcomes [37].

All procedures were performed by personnel with prior training in surgical and anesthetic techniques, thereby minimizing procedural trauma. As commonly practiced in human medicine [38], pre-oxygenation prior to anesthetic induction likely provided an oxygen reserve, which may offer advantages during the initiation of mechanical ventilation. During endotracheal intubation, magnification and a physiological monitoring system were employed to track temperature, peripheral capillary oxygen saturation (SpO_2_), heart rate (HR), and respiratory rate (RR) (Figure 2).

The use of appropriate materials further supported the model success. A custom platform was used to immobilize the rat’s head and maintain the oral opening in a fixed position, facilitating operator access (Figure 3). In our study, several technical challenges typically encountered in rodent models were overcome. The intervention was performed on the lower first molar, which was easily exposed by retracting the cheek and tongue using the same suction device. This allowed for manual confirmation of pulp exposure with an endodontic file and direct observation of pulp bleeding (Figure 3G,H).

Digital radiographic imaging (Figure 4) confirmed the retention of the filling material throughout the evaluation period. Finally, Hematoxylin and Eosin (H&E) staining (Figure 5) verified the presence of inflammation resulting from pulp exposure, as evidenced by the loss of tissue continuity compared to the negative control pulps, which remained intact and free of inflammatory infiltrate. These findings were further supported by the histopathological analysis (Table 1).

The examination of tissue samples stained with (H&E) from the MIPA, MIPZ, and NC groups, looked at after 3 h and 5 days, showed signs of pulp inflammation based on microscopic observation, as described in the Section 2 (Table 1).

Regarding the percentage of coronal area affected, in the MIPZ group at 3 h, 40% of the samples showed involvement between 1 and 30%, while 60% were affected between 31 and 60%. This distribution remained unchanged for five days. In the MIPA group at 3 h, 20% of the samples showed no involvement, while 80% were affected between 1 and 30%. At five days, a proportional increase in the affected area was observed. In the NC group, no involvement in the coronal area was observed at either time point.

As for the percentage of necrosis, variable scores were observed in the MIPZ group at 3 h; however, by day 5, all samples exhibited necrosis between 1 and 20%. In the MIPA group, 80% of the samples showed no necrosis at 3 h, but this proportion was reversed at 5 days, with necrosis present in 80% of the samples. In the NC group, necrosis was absent at both time points.

### 3.3. MIP Does Not Alter Animal Well-Being in the Early Stages

Three parameters associated with animal well-being were evaluated: body weight (g), food consumption (g) and water consumption (mL), measured one day before the procedure (i), twice on the day of the procedure, before the procedure (1), and three hours after the intervention (2) (Table 2).

The behavior of each parameter analyzed, weight, food and drinking water consumption, showed that there are no significant differences for MIPZ and MIPA, with respect to weight (*p* = 0.454) according to the Student’s *t*-test, indicating that the experimental units on the day before the intervention were the same, guaranteeing their homogeneity. The negative control group was not included in these inferential tests and is described only qualitatively to verify baseline behavior, in line with the 3Rs principle.

Regarding the distribution of water (WCi-1) and food consumption (FCi-1) between the previous day and the day of the procedure, no significant differences were observed between the treatment groups MIPZ and MIPA (*p* = 0.707, *p* = 0.355) as determined by the Student’s *t*-test for water consumption and the Mann–Whitney U test for food consumption as shown in Table 3 (*p* = 0.56). The variance homogeneity was further corroborated by Levene’s test (*p* > 0.05) and is illustrated in Figure 6.

Median values for all variables measured three hours post-procedure, body weight (*p* = 0.456), food consumption (*p* = 0.73), and water consumption (*p* = 0.241) did not differ significantly between the experimental groups (MIPA and MIPZ), as determined by the Student’s *t*-test for weight and Mann–Whitney U test for food and water consumption indicating that the experimental units on the day before the intervention were the same, guaranteeing their homogeneity. The negative control group remains excluded from inferential analysis and is reported descriptively according to the 3Rs principle to reduce animal use.

Taken together, these findings indicate that the MIP procedure and the treatments applied in each group did not produce immediate, detectable effects on animal well-being within the first three hours following intervention. Specifically, the parameters of body weight, food consumption, and water consumption remained stable across all treatment groups, suggesting that the different treatments did not significantly alter these indicators of well-being in the short term.

Furthermore, Levene’s test confirmed the homogeneity of variances (*p* > 0.05), and the initial conditions were shown to be evenly distributed across all groups (Table 3). These results validate the randomization process and confirm the baseline equivalence of the experimental units.

### 3.4. Animal Well-Being Is Preserved Following Five Days of MIP Induction

During the five days after the treatment, a minor initial decrease in body weight was observed in the MIPA and MIPZ groups, followed by progressive recovery beginning on day 3 (Figure 6).

In contrast, the negative control group showed a continuous trend of weight gain. No statistically significant differences were detected between treatments groups, and the coefficients of variation remained low (6.2–8.5%) (Table 4), indicating that weight-related measures were not negatively impacted by the procedure.

The Shapiro–Wilk normality test confirmed that body weight was normally distributed at each observation point, regardless of the treatment (*p* = 0.616, *p* = 0.607, *p* = 0.72, *p* = 0.799). Additionally, Levene’s test (*p* > 0.05) for homoscedasticity (*p* = 0.991) confirmed a constant variance across groups, supporting the assumption of data homogeneity across all time points (W1, W2, W3, W6) and treatment conditions (MIPA and MIPZ) (Table 5).

The analysis of food consumption over specific periods showed a cyclical pattern characterized by an initial increase on the day of the treatment, followed by a decrease between days 2 and 3, and subsequent recovery (Figure 7A), which can be verified with the coefficients of variation (6.6%; 22%; 8.9%; 11.7%) for the MIPZ group and (9.2%; 17.7%; 9.5%; 16.5%) for the MIPA group (Table 5). The trend at all times remained consistent across all experimental groups. The most significant coefficients of variation (CV) were observed between days 1 and 2 (FC1-2), with values of 22% and 17.7% for the MIPZ and MIPA groups, respectively. In the negative control group, the maximum CV was 21% at the FC2-3 interval.

Water consumption showed a slight decline between days 1 and 2, followed by stabilization through day 5 (Figure 7B). Variability in water intake was more pronounced, with the highest CVs recorded as follows: 22.8% for MIPZ on the final day of evaluation, 44.4% for MIPA on the day of the procedure, and 41.5% for the negative control group on the day prior to the procedure.

Additionally, weight gain (+) or loss (-) ratios between time intervals were calculated using Equation (1), and the total percentage of gain or loss was determined as an absolute value using Equation (2):(1)$$RGL_{i}=\left(\frac{X_{i}-X_{i-1}}{X_{i}}\right)\times 100\%$$(2)$$\Delta =\sum RGL_{k}$$

As shown in Table 6, the greatest total body weight gain occurred between the final two time points evaluated, with a consistent upward trend observed across all treatment groups. This indicates a general recovery and stabilization of physiological parameters following the procedure.

On the other hand, water consumption showed a decrease, particularly in the MIPA group, which showed the highest reduction in percentage. Regarding food consumption, the total gain exceeded the loss of 31.7 g (Table 6).

Additionally, the overall weight gain (∆ Overall) was calculated as the difference between the final and initial body weights, as defined by Equation (3):(3)$$\Delta \;Overall=∥W_{f}-W_{i=0}∥$$

This metric provides an accumulative measure of weight change throughout the experimental timeline and serves as an additional indicator of animal well-being.

In percentage terms, the distribution of weight change prior to treatment application was 57% for the MIPA group, 36% for MIPZ, and 7% for the control group (Table 6). Regarding total weight loss (9.7 g) during this same period, the highest loss was observed in the MIPA group (5.2 g).

Moreover, the rate of daily weight gain or loss was calculated using Equation (4):(4)$$r_{GPW}=\frac{W_{f}-W_{i=0;1}}{n}\;\;\;\left[g/\mathrm{days}\right]$$

This parameter provides insight into the speed at which animals recovered or lost weight over time, offering an additional layer of analysis for evaluating the impact of the treatments on animal well-being.

By day five, the trend of weight gain surpassing weight loss was maintained, with a net increase of 11.25% over the fourth day (Table 7). MIPA exhibited the greatest total weight gain (49.2 g), followed by MIPZ (31 g) and the control group (6 g) among the treatment groups. These results further support the observation that, despite initial fluctuations, animals in all groups recovered steadily, with the most pronounced gains observed in the treated groups.

### 3.5. The Behavior of the Animals Is Unchanged During Monitoring

Clinical evaluation using the standardized well-being assessment format revealed that nearly all animals showed no alterations in the evaluated parameters. The only exception was mild passivity, observed in 50% of the animals during the initial days post-procedure; however, this sign did not persist beyond day 5.

Representative images illustrating the preserved well-being of the rats following mechanical induction of pulpitis (MIP) are shown in Figure 8. These include the absence of piloerection, normal respiratory rate patterns, upright posture without signs of slouching, absence of ocular discharge, typical resting or exploratory behavior, escape response upon handling, social interaction with peers, a well-maintained coat without skin lesions or alopecia, and evidence of normal food and water consumption (Figure 8A–F).

These observations support the conclusion that the MIP procedure did not result in sustained behavioral or clinical signs of distress, further confirming the preservation of animal well-being throughout the experimental period.

## 4. Discussion

A model of mechanically induced pulpitis, sealed with amalgam or zinc phosphate, was established in this study without compromising the well-being of the rats during the evaluation period. In contrast to other models described in the literature, such as those requiring lipopolysaccharide (LPS) stimulation, caries-induction, laser application, or transgenic modifications, some of which involve a carbohydrate-rich diet that may lead to obesity and diabetes, the inoculation of one or more bacterial strains to promote a diverse microbial environment, and even surgical recession of salivary glands to reduce salivary flow and facilitate caries development [40], the model presented here offers a simpler model that prioritizes animal well-being while maintaining scientific relevance. It closely resembles clinical scenarios of pulpitis caused by mechanical injury, often of iatrogenic causes in humans, and allows for the study of pulpal inflammation without inducing stress or compromising animal well-being.

Importantly, this model also enables the exploration of therapeutic strategies for the treatment of pulpitis in future research, with the potential to preserve pulp tissue and maintain tooth vitality in a clinical setting. Understanding the inflammatory process and evaluating promising anti-inflammatory, restorative or regenerative agents such as drugs requires the use of experimental animals. In this context, the implementation of standardized evaluation systems, carried out by trained experienced personnel can significantly reduce the severity of the procedures. These systems rely on simple, objective measurements to detect the onset and progression of pain, suffering, and distress in animals subjected to scientific experimentation [41].

Reducing or avoiding pain and stress in rodents used for biomedical research is both an ethical obligation and a regulatory requirement [42], which is crucial to ensure valid outcomes in studies of inflammation, pain or immunology [43]. The use of animal models must be thoroughly justified, and experimental procedures must be conducted with the highest standards to ensure animal well-being. Pain relief is commonly accomplished by the administration of analgesic agents, such as opioids and nonsteroidal anti-inflammatory drugs [42]. However, the use of these agents can alter the cellular and molecular responses to inflammation. In particular, they may specifically inhibit proteins involved in a role in the synthesis of inflammatory mediators, potentially altering experimental outcomes due to their immunomodulatory properties or synergistic effects [44,45,46,47,48,49,50] and failure to recognize postoperative pain [51,52,53,54,55].

Based on the above, and considering the results obtained in this study, the administration of analgesics was deemed unnecessary in the developed model. The evaluation of pain, discomfort or distress indicators demonstrated that this sealed model of mechanically induced pulpitis can be used to investigate pain mechanisms without compromising animal well-being. The presence of pulp injury was confirmed by visible bleeding, consistent with previous reports in the literature [21]. Given that pulp inflammation is known to cause pain and potentially affect animal well-being [21], behavioral changes were monitored over time. Prior to the main study, a pilot experiment was conducted involving ten rats with mechanically induced pulpitis and sealed with amalgam or zinc phosphate. These animals were evaluated through video recordings by two independent, trained and calibrated observers who were blinded to treatment allocation, using the *Rifai* [32], *Grimace* [33], and *Frecknell* [34] scales.

Remarkably, all parameters associated with dental pain, such as facial expression and abnormal head or body movements, scored zero across all animals. These findings support the conclusion that the model does not induce observable pain-related behaviors and may be considered a refined approach for studying pulpitis while preserving animal well-being. Although the scoring system allows for the classification of behaviors as mild, moderate, or substantially severe, no pain-related behaviors (e.g., head shaking, facial grimacing) were identified upon consensus review, confirming the original score of “0” for all pain parameters. It is important to note that the pain scales and scoring system used in this study are discrete, which may limit the detection of subtle changes in pain behavior.

The results of the pilot study supported the expansion of the sample size and the implementation of structured animal well-being monitoring and evaluation formats. These formats included predefined thresholds for clinical and behavioral signs, along with scoring systems to determine a humane endpoint. The structure of these monitoring sheets aligns with the guidelines proposed by the Federation of European Laboratory Animal Science Associations (FELASA) in 1994 [35] and incorporates subsequent updates and refinements [36].

The findings of this study demonstrated no statistically significant variations in body weight, food consumption, or water consumption during the evaluations period, suggesting that the model implemented does not adversely affect the well-being of the animals involved in the procedure, possibly due to the application of refinement strategies [56]. Although baseline weights were comparable, formal stratified randomization by weight was not performed and could be incorporated in future studies.

Some of the clinical signs observed, such as the mild passivity recorder three hours after the procedure, may be attributed to the effects of isoflurane, the anesthetic used in this study. Isoflurane is known to reduce blood pressure and systemic vascular resistance [57]; due to its effects on the central nervous system, it can induce drowsiness or reduce motor activity for a few hours after the suspension of the anesthesia and influence the slight decrease in food consumption observed on the day of the procedure.

Male Lewis rats were selected to ensure greater homogeneity in the well-being assessments, particularly regarding food intake, a key variable in this study, given their larger body size compared to females. Additionally, males exhibited calmer behavior during handling, facilitating behavioral evaluations. Although few studies have addressed sex-based differences in pain responses, such as evidence by Zorczeniewska, 2017 [58].

Models for inducing pulpitis in rodents have technical challenges, primarily due to their small anatomical size [5]; to address limitations, several refinement strategies were implemented in the present study. First, rats were utilized; although the morphology of the molar is predominantly analogous to that of mice, the linear dimensions are twice as large [59], facilitating surgical access. Furthermore, in contrast to incisors which exhibit continuous eruption due to open apices [60], the molar was employed as recognized by the scientific community; lower molars are widely regarded as the most reliable “dental proxy” for modeling human dental conditions [59].

The exposure of the pulp was unilateral, which is different from the bilateral (contralateral) molar intervention used in several other studies. On the other hand, the procedure was improved by ensuring adequate oral access through physical immobilization of the rat’s head. This facilitated accurate access using a ¼ round bur activated with medicinal-grade nitrogen (N_2_) instead of compressed air, thus minimizing the risk of introducing airborne pollutants into the pulp chamber.

In rats, endotracheal intubation, a prerequisite for artificial pulmonary ventilation [61], is very challenging due to its small size, fast breathing rate, narrow oral cavity, and high position of the glottis, making it difficult to expose the glottis during intubation [62]. To refine this technique, training was first conducted using cadaver specimens; specialized intubation instruments designed for rats were employed, along with a light source and a 3.5× surgical telescope, which enables clear visualization of the movement of vocal cords. This approach resulted in a technique that is safe, reproducible, and relatively easy to perform, while minimizing the risk of mortality and perioperative complications such as laryngeal edema, tracheal perforation, hemorrhage, or airway obstruction due to stimulated secretions [63]. Rigorous postoperative monitoring was performed for 30 min following anesthetic recovery. This involved the use of external heat sources to prevent hypothermia-associated discomfort [37]. Precise control of postoperative recovery in laboratory animals is crucial to ensure the optimal standards of animal well-being and care [64]. During this period, the restoration of vital signs including normal respiration, as well as capacity to stand up and walk within the recovery area, was observed. Upon the fulfillment of these conditions, the rats were returned to their housing, where social interaction with their cage mates significantly contributed to the restoration of their physical condition [65].

The model used in this study is well controlled. Following exposure of the pulp, the cavity was sealed with amalgam or zinc phosphate, effectively preventing the entrance of materials or microorganisms from the oral environment. Throughout the experimental evaluation period, the sealing material remained in place in the treated teeth, ensuring the stability of the experimental conditions and minimizing external contamination. The procedures were controlled for saliva contamination [66] through the implementation of the head fixation (Figure 3B), oral opening fixation (Figure 3G–L), tongue retraction, facial muscle retraction (Figure 3F), disinfection and drying of the peridental area, as well as tooth access as described in Materials and Methods. The cavity size was standardized and consistently prepared across all specimens; previous research has shown that cavity size can influence the pulp healing response following capping procedures [67].

This approach induces a state of chronic inflammation, initially triggered by mechanical trauma and potentially exacerbated by the cytotoxic effect of amalgam [39,68] or the acidity of zinc phosphate [25,27]. The use of amalgam, especially when it is in direct contact with the pulp, may cause prolonged inflammation due to the release of metal ions; these ions can induce oxidative stress and chronic inflammation even at non-cytotoxic concentrations [26]. This situation could lead to clinical signs at the pulp level, increasing the risk of tissue degradation, fibrous tissue formation or necrosis, depending on how intense and long-lasting the chemical stimulus is [24]. Acid components (low pH) of zinc phosphate directly can cause pulp pain, inflammation, or potentially necrosis [25], because this may stimulate odontoblast activity and increase the release of proinflammatory mediators Furthermore, increased dentinal tubule permeability could facilitate acid perfusion, triggering a reactive inflammatory response in odontoblasts and dental pulp [26]. These previous studies were conducted in humans and non-human primates; therefore, this study is the first to propose the use of traditionally employed dental materials that can induce non-pathological inflammatory effects due to their composition in a rat model.

Additionally, this model does not induce pain, as evidenced by the observed conditions of well-being following MIP. Although occlusal trauma could not be conventionally confirmed due to the animal being intubated under anesthesia, this type of anesthesia provided the advantage of allowing for sufficient time to perform the obturation optimally, aiming to preserve the occlusal table and maintain the characteristic morphology of the first molar. Consequently, occlusal or mechanical trauma or pulpitis could be ruled out [69].

In this study, pulp inflammation was evidenced both clinically and histopathologically. Clinically, it was observed through the presence of local inflammation within the coronal pulp, allowing for the evaluation of the true inflammatory status of the pulp tissue [40]. Histopathologically, the inflammatory status following MIPA and MIPZ was described using two parameters: the percentage of the coronal area affected by inflammatory infiltrate and the percentage of necrosis. The MIPZ group showed a stable distribution of inflammatory involvement over time, while the MIPA group exhibited a progressive increase in the affected area with tendency toward greater necrosis, as also reported in previous studies [24,25,26].

These findings are consistent with previous studies reporting that nociceptive activation does not occur in sealed cavities [20]. On the other hand, the introduction of bacteria into the pulp chamber can lead to irreversible pulpitis, which impairs the tissue’s ability to heal spontaneously and ultimately results in pulp necrosis and the death of tissue [5,70]. This bacterial invasion promotes the nociceptive response [71], highlighting the importance of preserving a sealed environment to isolate the mechanical and chemical components of inflammation from microbial factors.

The mechanisms that cause inflammation of the pulp are not yet fully understood [72], which emphasizes the importance of developing experimental models of pulpitis in animals. Such models are essential for investigating the pathophysiology of the disease and for evaluating potential therapeutic strategies aimed at reducing its clinical and economic burden [71]; among these, models that allow for the possibility of pulp healing traditionally referred to as reversible pulpitis are particularly valuable [5].

The model established in this study, consistent with previous research, confirmed pulp injury through the observation of bleeding [21]; additionally, H&E staining revealed a controlled inflammatory response, which progressed to the middle third of the coronal pulp after five days of evaluation. Inflammatory processes such as those observed here, in which there is no evidence of nociceptive response, are considered under a translational framework that equates the experience of pain across species, mouse = rat = human [73]. This may be comparable to certain chronic irreversible conditions in humans that can remain asymptomatic [74] and may help explain the preservation of animal well-being observed in this study.

To better replicate the complexity of human disease, some studies have introduced inflammation-inducing agents such as lipopolysaccharide (LPS) directly into the pulp chamber [72], or tested compounds with a therapeutic potential, such as lipoic acid [60]. The model developed in this study allows for the creation of a space between the pulp and the sealing material, that can be used for the introduction of molecules that represent in a complementary way the development of targeted delivery of molecules. This feature enables the simulation of the progression of disease or the localized application of immunomodulatory compounds with therapeutic potential, thereby expanding the model’s utility for both mechanistic and translational research.

Despite certain limitations such as the restricted oral opening in a small animal model like the rat, which hinders the use of essential rotary instruments (e.g., high-speed handpiece with the dental drill, suction device, amalgam carrier and microcondenser for endodontic surgery), this study demonstrated that the use of a head fixation platform facilitates the procedure without compromising the airway. Therefore, for future research, the use of adult rats weighing between 350 and 400 g has the advantage of providing an oral aperture wide enough to allow for the introduction of all necessary experimental procedures without compromising animal well-being. In this study, our model focuses on inflammation rather than repair. We selected adult rats based on evidence that they exhibit a more prominent inflammatory response, which may resemble that of adult humans [15,75], in contrast to young rats, which are more commonly associated with regenerative or reparative responses [76,77,78]. This model was developed to enable future testing of drugs with therapeutic potential at a pulp level. Given the priority placed on establishing the model, the scope of this study did not include testing the effect of immunomodulatory molecules, except in a pilot phase where they were used to induce or assess increased inflammation and signs related to pain.

Overall, the findings of this study support the implementation of a controlled, live, and reproducible model for evaluating therapeutic candidates aimed at managing pulp inflammation. Nevertheless, the adoption of this approach must be accompanied by a thorough assessment of the potential pain induced in experimental animals.

## 5. Conclusions

A model of mechanically induced pulpitis was developed in molar Lewis rats, sealed with either amalgam or zinc phosphate without compromising animal well-being in this study. This model provides a valuable tool for investigating therapeutic candidates and elucidating the underlying mechanisms of pulp inflammation.

## Figures and Tables

**Figure 1 biomedicines-13-01925-f001:**
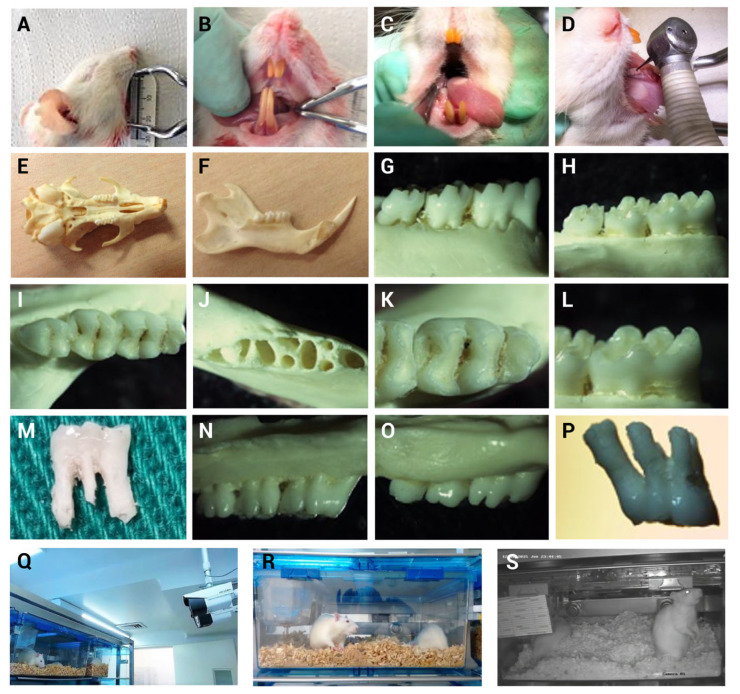
Surgical skills were developed using discarded specimens and morphological characteristics of Lewis rat molars and rats were monitored following MIP induction throughout the pilot study. (**A**) Oral cavity opening. (**B**) Dentition characteristics. (**C**) Visualization of the lower first molar. (**D**) Verification of handpiece and bur access to the mesial fossa of the lower first molar. (**E**) Maxilla (upper jaw). (**F**) Mandible (lower jaw). (**G**) Buccal view of lower molars. (**H**) Lingual view of lower molars. (**I**) Occlusal view of lower molars. (**J**) Lower molar alveoli. (**K**) Occlusal surface of the lower first molar. (**L**) Cusps of the lower first molar. (**M**) Lower first molar. (**N**) Buccal view of upper molars. (**O**) Palatal view of upper molars. (**P**) Upper first molar. (**Q**–**S**) Rat behavior within their housing cages was continuously monitored through 24/7 video recording for seven days following MIP induction.

**Figure 2 biomedicines-13-01925-f002:**
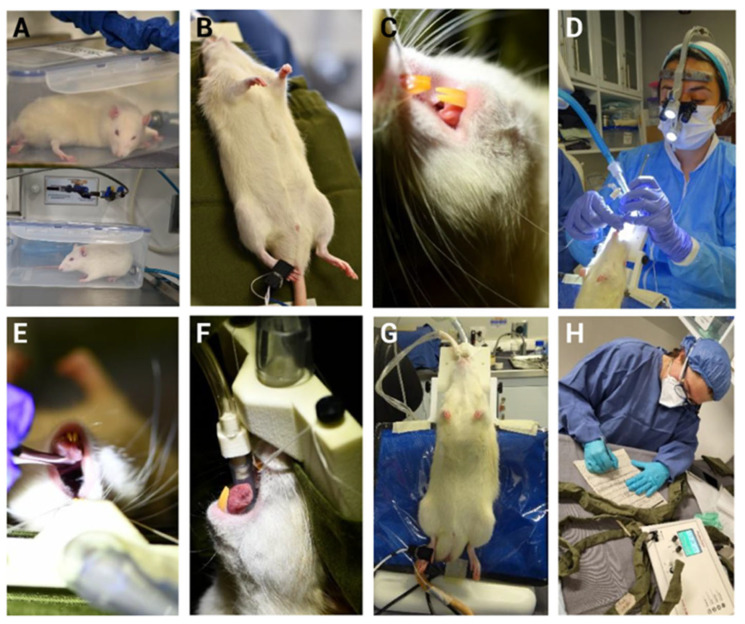
Anesthetic induction, maintenance, and endotracheal intubation. Representative images are shown: (**A**) Anesthetic induction with 3% isoflurane. (**B**) Anesthetic maintenance with 3% isoflurane via nasal mask and placement of sensors for signal monitoring of paws, including heart rate (HR), respiratory rate (RR), peripheral capillary oxygen saturation (SpO_2_), and body temperature. (**C**) Animal fixation using upper incisors. (**D**) Trained personnel performing endotracheal intubation with a 3.5× magnifying telescope. (**E**) Oral cavity opening for localization of vocal cords. (**F**) Endotracheal intubation. (**G**) Anesthetic maintenance via endotracheal intubation. (**H**) Vital signs were continuously monitored and recorded throughout the procedure using the PhysioSuite system.

**Figure 3 biomedicines-13-01925-f003:**
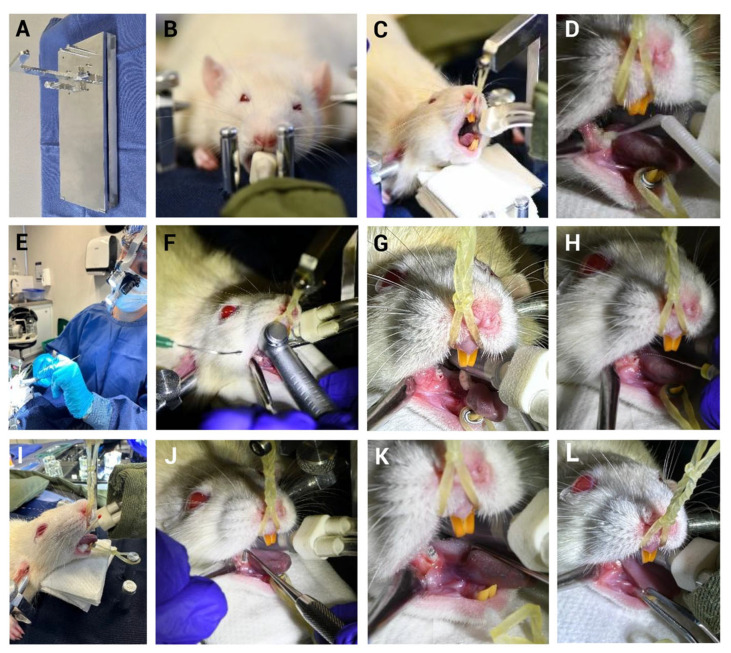
Mechanically induced pulpitis sealed in the lower first molar of a Lewis rat. Representative images are shown: (**A**) Fixation platform. (**B**) Skull stabilization using auricular bars. (**C**) Placement of orthodontic elastics from upper incisors to the platform. (**D**) Fixation of lower incisors using elastic bands and ligatures; molar cleaned with 2% chlorhexidine. (**E**) Endodontist personnel using a light and 3.5× magnification system. (**F**) Occlusal surface opening at the mesial fossa of the lower first molar using a high-speed handpiece with saline irrigation. (**G**) Pulp exposure was identified following the observation of localized bleeding. (**H**) Confirmation of pulp exposure using a #15 endodontic file. (**I**) Drying of the cavity with paper points. (**J**) Condensation of sealing material. (**K**) Cavity sealing with amalgam. (**L**) Cavity sealing with zinc phosphate cement.

**Figure 4 biomedicines-13-01925-f004:**
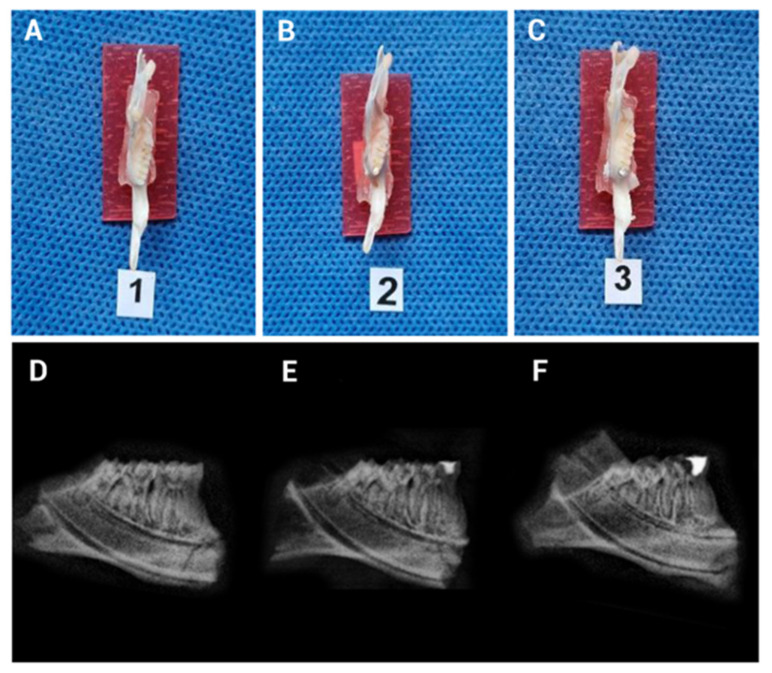
Representative clinical and radiographic images of the mandible of a Lewis rat following mandibular dissection. (**A**) Image on the right mandible without mechanically induced pulpitis in the lower first molar. (**B**) Image showing mechanically induced pulpitis (MIP) with the cavity sealed using zinc phosphate cement. (**C**) Image showing MIP with the cavity sealed using amalgam. (**D**–**F**) Digital radiographic images corresponding to the described samples.

**Figure 5 biomedicines-13-01925-f005:**
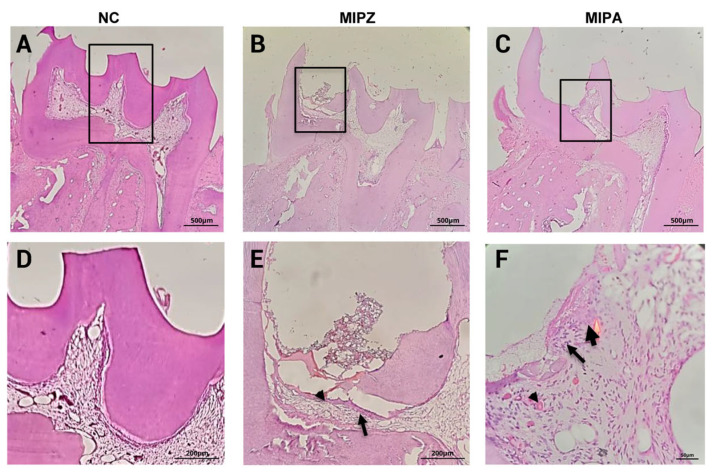
Histological results (Hematoxylin and Eosin) H&E staining of rat pulp tissue without stimulation and following sealed pulpitis with amalgam or zinc phosphate after 3 h. (**A**–**C**) (magnification, 5× at 500 μm). (**A**) Negative control (NC) pulp tissue, no inflammatory infiltrate is observed. (**B**) Following pulp exposure sealed with zinc phosphate (MIPZ). (**C**) Following pulp exposure sealed with amalgam (MIPA), where inflammatory infiltrate is observed in approximately 30% of the coronal pulp. (**D**,**E**) High-magnification (10× at 200 μm) and (**F**) (magnification, 40× at 50 μm), views of the boxed areas are shown in (**A**–**C**), respectively. The triangles in (**E**,**F**) indicate blood vessels containing red blood cells, and arrows point to extravasated neutrophils. All these micrographs were taken with a Zeiss Axiolab 5 microscope.

**Figure 6 biomedicines-13-01925-f006:**
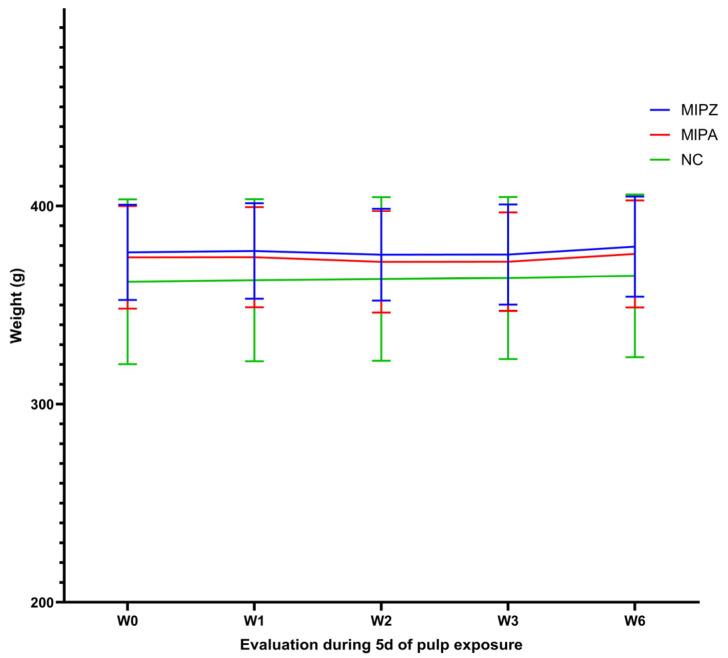
Evaluation of body weight in Lewis rats over five days following the MIP procedure. The figure shows the body weight on the rats at five different time points: W0: before the procedure; W1: on the day of the procedure; W2: two days after; W3: three days after; and W6: five days after the procedure. This evaluation was conducted across the three study groups. The mean and standard deviation of body weight in grams for each group are presented for each time point.

**Figure 7 biomedicines-13-01925-f007:**
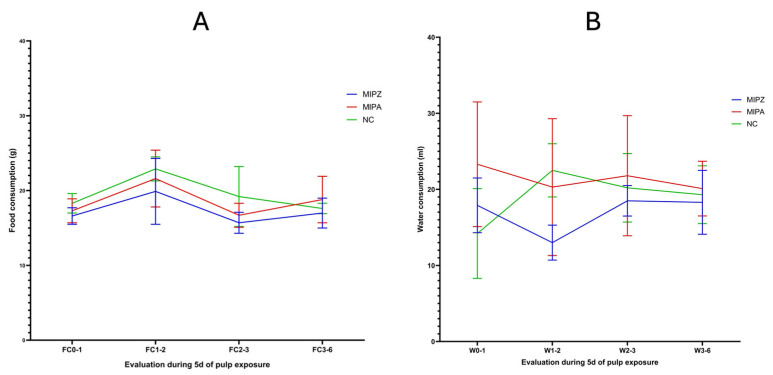
Evaluation of food and water consumption over five days following the MIP procedure. The figure shows (**A**) food consumption and (**B**) water consumption on the rats at four different time points: (**A**) Food consumption, FC0-1: between the day before and the day of MIP; FC1-2: between the day of and the day after MIP; FC2-3: between day three and day two after MIP; FC3-6: between the last day of MIP evaluation and the previous day. (**B**) Water consumption, W0-1: between the day before and the day of MIP; W1-2: between the day of and the day after MIP; W2-3: between day three and day two after MIP; W3-6: between the last day of MIP evaluation and the previous day. This evaluation was conducted across the three study groups. The mean and standard deviation of total food consumption in grams and water consumption in mL by group are presented for each assessment point.

**Figure 8 biomedicines-13-01925-f008:**
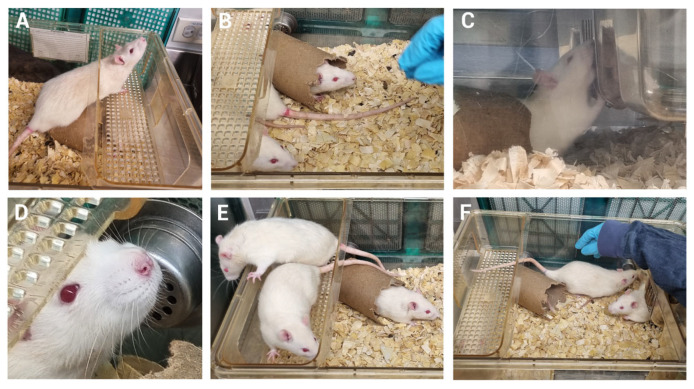
Images of animal behaviors not associated with pain. (**A**) Absence of piloerection. (**B**) Normal elicited behavior patterns without passivity. (**C**) Normal posture without hunching and food consumption. (**D**) No ocular discharge vs. presence of discharge (close-up image near the eyes). (**E**) Normal posture (sleeping, exploring, digging) with absence of immobility and high level of social interaction among cage mates. (**F**) Escape response to handling. In general, normal fur without skin lesions or alopecia was observed in all animals involved in this study.

**Table 1 biomedicines-13-01925-t001:** Histopathological analysis of Hematoxylin and Eosin (H&E) stained samples at a 3-h period and 5 days in MIPZ, MIPA, and NC groups.

	3 h			5 Days		
Parameter	NC (n = 2)	MIPZ (n = 10)	MIPA (n = 10)	NC (n = 2)	MIPZ (n = 10)	MIPA (n = 10)
% Coronal area affected by inflammatory infiltrate	1 = 100%	2 = 40%	1 = 20%	1 = 100%	2 = 40%	2 = 40%
		3 = 60%	2 = 80%		3 = 60%	3 = 60%
% Necrosis	1 = 100%	1 = 20%	1 = 80%	1 = 100%	2 = 60%	1 = 20%
		2 = 40%				2 = 40%
		3 = 20%	3 = 20%		3 = 40%	3 = 40%
		4 = 20%				

The numbers in the table represent the evaluation parameter scores. The percentage (%) represents the relative frequency of each score within the study groups. % Coronal area affected by inflammatory infiltrate: Score 1 = Absent; Score 2 = 1–30%; Score 3 = 31–60%; Score 4 = 61–100%. % Necrosis: Score 1 = Absent; Score 2 = 1–10%; Score 3 = 11–20%; Score 4 => 20%.

**Table 2 biomedicines-13-01925-t002:** Average values of parameters associated with well-being throughout the trial over a 3-h period.

	Treatments		
Parameter	MIPA $\mu \left(\sigma \right)$	MIPZ $\mu \left(\sigma \right)$	NC $\mu \left(\sigma \right)$
n	(n = 10)	(n = 10)	(n = 2)
Weight [g]	382.5 (33.56)	394.1 (32.8)	316 (7.36)
Food consumption [g]	9.9 (5.37)	10.4 (5.7)	9.97 (5.75)
Water consumption [39]	9.6 (7.04)	9.6 (6.7)	8.5 (6.6)

*μ*: media; *σ*: standard deviation.

**Table 3 biomedicines-13-01925-t003:** Descriptive statistics of the factors at different points in time of the experiment.

Treatment	Parameters	Wi (g)	W1 (g)	W2 (g)	FCi-1	FC 1-2	WCi-1	WC 1-2
**MIPZ**	*μ*	393.9	394.6	393.9	15.6	5.2	15.9	3.2
	*σ*	34.1	33.9	34.0	2.1	1.8	1.8	0.6
**MIPA**	*μ*	382	383.2	382.3	14.8	5.0	16.3	2.9
	*σ*	35.6	34.6	34.1	2.2	1.7	2.2	0.4
**NC**	*μ*	312.6	315.5	316.0	14.8	5.2	14.2	2.8
	*σ*	11.5	9.9	9.9	2.6	1.0	1.2	0.4

Wi (g): weight from the day before the procedure; W1 (g): weight before the procedure; W2 (g): weight after the procedure. FCi-1: food consumption from the day before the procedure combined with consumption before the procedure on the day of the procedure; FC 1-2: food consumption before the procedure combined with consumption 3 h after the procedure. WCi-1: water consumption from the day before the procedure combined with consumption before the procedure on the day of the procedure; WC 1-2: water consumption before the procedure combined with consumption 3 h after of the procedure. *μ*: media; *σ*: standard deviation.

**Table 4 biomedicines-13-01925-t004:** Average values of parameters associated with well-being throughout the trial over a 5-day period (General features of animal database, n = 22).

Parameter	Treatments		
	MIPA	MIPZ	NC
**Weight [g]**	(n = 10)	(n = 10)	(n = 2)
*μ* (*σ*)	373 (24.7)	376.9 (23.4)	363.2 (30.7)
Mean confidence interval	[33]	[370.2–383.5]	[34]
Coefficient of variation	6.60%	6.20%	8.50%
**Food consumption [g]**	(n = 10)	(n = 10)	(n = 2)
*μ* (*σ*)	18.6 (3.23)	17.3 (2.95)	19.4 (2.78)
Mean confidence interval	[17.6–19.6]	[16.3–18.3]	[17.1–21.8]
Coefficient of variation	17.40%	17%	14.30%
**Water consumption** [39]	(n = 10)	(n = 10)	(n = 2)
*μ* (*σ*)	21.4 (7.3)	16.9 (3.8)	19 (4.7)
Mean confidence interval	[19–23.7]	[15.7–18.2]	[15.1–23]
Coefficient of variation	34.1%	22.5%	24.7%

*μ*: media; *σ*: standard deviation.

**Table 5 biomedicines-13-01925-t005:** Descriptive statistics of parameters associated with well-being throughout the trial over a 5-day period (General features of animal database, n = 22).

Descriptive Statistics													
MIPZ	Wi	W1	W2	W3	W6	FCi-1	FC1-2	FC2-3	FC3-6	WCi-1	WC1-2	WC2-3	WC3-6
*μ*	376.8	377.3	375.4	375.5	379.5	16.6	19.9	15.7	17.0	17.9	13.0	18.5	18.3
σ	24.0	24.1	23.2	25.3	25.3	1.1	4.4	1.4	2.0	3.6	2.3	2.0	4.2
MIPA	Wi	W1	W2	W3	W6	FCi-1	FC1-2	FC2-3	FC3-6	WCi-1	WC1-2	WC2-3	WC3-6
*μ*	371.4	374.2	371.8	371.9	375.8	17.3	21.6	16.7	18.8	23.3	20.3	21.8	20.1
σ	25.9	25.3	25.6	24.9	27.0	1.6	3.8	1.6	3.1	8.2	9.0	7.9	3.6
NC	Wi	W1	W2	W3	W6	FCi-1	FC1-2	FC2-3	FC3-6	WCi-1	WC1-2	WC2-3	WC3-6
*μ*	361.8	362.6	363.2	363.7	364.8	18.3	22.9	19.2	17.6	14.2	22.5	20.2	19.3
σ	41.6	40.9	41.3	40.9	41.1	1.3	1.6	4.0	0.7	5.9	3.5	4.5	3.8

Wi (g): weight from the day before the procedure; W1 (g): weight before the procedure; W2 (g): weight one day after the procedure; W3: weight two days after the procedure; W6: weight five days after the procedure. FCi-1: food consumption from the day before the procedure combined with consumption before the procedure on the day of the procedure; FC 1-2: food consumption from the day of the procedure combined with consumption one day after the procedure; FC2-3: food consumption from one day of the procedure combined with consumption two days after the procedure; FC3-6: food consumption from the second day of the procedure combined with consumption five days after the procedure. WCi-1: water consumption from the day before the procedure combined with consumption before the procedure on the day of the procedure; WC1-2: water consumption from the day of the procedure combined with consumption one day after the procedure; WC2-3: water consumption from one day of the procedure combined with consumption two days after the procedure; WC3-6: water consumption from the second day of the procedure combined with consumption five days after the procedure *μ*: media; *σ*: standard deviation.

**Table 6 biomedicines-13-01925-t006:** Total percentage of weight gain or loss. Values were obtained using Equations (1) and (2).

T3a Weight			
Total percentage of gain or loss in each period (Δ)	MIPZ % (g)	MIPA % (g)	NC % (g)
Overall growth ($\Delta_{Wi}$)	4.1 (15.4)	8.7 (31.5)	0.5 (1.7)
Overall growth ($\Delta_{W1}$)	0.85 (2.8)	0 (0)	0.32 (1.2)
Overall growth ($\Delta_{W2}$)	4.11 (16)	3.14 (11.4)	0.29 (1.0)
Overall growth ($\Delta_{W3}$)	10.6 (39.7)	10.1 (38.7)	0.57 (2.1)
Overall decrease ($\Delta_{Wi}$)	2.8 (10.6)	1.1 (3.9)	0 (0)
Overall decrease ($\Delta_{W1}$)	5.6 (21.3)	6.4 (23.9)	0 (0)
Overall decrease $(\Delta_{W2}$)	4.2 (15.5)	2.6 (9.8)	0 (0)
Overall decrease ($\Delta_{W3}$)	0 (0)	0 (0)	0 (0)
T3b Food consumption	MIPZ % (g)	MIPA % (g)	NC % (g)
Overall growth ($\Delta_{Fi}$)	212.7 (36.2)	262.5 (44)	50.4 (9.2)
Overall growth ($\Delta_{F1}$)	16.7 (2.5)	0 (0)	0 (0)
Overall growth ($\Delta_{F2}$)	107.9 (16.4)	141.6 (24)	11.04 (1.8)
Overall decrease ($\Delta_{F1}$)	14.3 (2.5)	9.3 (1.8)	0 (0)
Overall decrease ($\Delta_{F1}$)	189.4 (44.4)	202.9 (49)	33.2 (7.4)
Overall decrease $(\Delta_{F2}$)	22.2 (3.4)	15.4 (2.2)	22.3 (4.9)
T3c Water consumption	MIPZ % (mL)	MIPA % (mL)	NC % (mL)
Overall growth ($\Delta_{Hi}$)	0 (0)	298 (49.8)	159.3 (16.7)
Overall growth ($\Delta_{H1}$)	485.8 (55.3)	436 (64.2)	16.5 (3.3)
Overall growth ($\Delta_{H2}$)	163.6 (26)	140.2 (22.1)	29.4 (5)
Overall decrease ($\Delta_{H1}$)	264.7 (49)	270 (86)	0 (0)
Overall decrease ($\Delta_{H1}$)	0 (0)	149.5 (49.8)	32 (8)
Overall decrease $(\Delta_{H2}$)	138.5 (27.7)	125.2 (38.4)	29.5 (6.7)

**Table 7 biomedicines-13-01925-t007:** Average weight gain and loss in absolute values and weight gain/loss index over the 5-day evaluation period.

	∆ Overall GPW0	∆ Overall PPW0	Index GPW0	Index PPW0
MIPZ	31	4.5	6.2	0.9
MIPA	49.2	5.2	9.84	1.04
NC	6	0	1.2	0
Total	86.2	9.7	17.24	1.94

## Data Availability

The original contributions presented in this study are included in the article. Further inquiries can be directed to the corresponding authors.

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
