# Peer review of "Mechanically Induced Pulpitis: A Rat Model That Preserves Animal Well-Being"

_biomedicines, 2025, doi:10.3390/biomedicines13081925_

Round 1
Reviewer 1 Report
Comments and Suggestions for Authors
Reviewer commnet,
This manuscript presents a postoperative behavioral evaluation using a mechanically induced pulpitis model. While the study falls within the scope of Biomedicine, there are several concerns that should be addressed to improve the scientific rigor and clarity of the manuscript.
Regarding the materials chosen, amalgam and zinc phosphate cement are generally not considered biocompatible with the dental pulp. They are not treated as direct pulp cappping materials.
The authors should clearly justify their use in this study.
The authors applied capsaicin to the exposed pulp, but the manuscript lacks essential details such as the concentration used, the volume applied, the physicochemical properties of the solution, and the specific application method.
Add detail information for reproducibility.
The cavity on mandibular molar was prepared mechanically and then exposed pulp was capped with materials. How was saliva contamination controlled during the procedure?
The authors should describe the measures taken to minimize contamination.
Previous studies have emphasized the importance of strict contamination control.
(e.g., doi: 10.1007/s00784-018-2374-5)
After restoring the cavity with amalgam or zinc phosphate cement, how was occlusal adjustment verified?
Describe the method used to confirm that occlusal trauma was not introduced during the procedure. I am concerned that it is not possible to determine whether postoperative pain and behavioral changes are due to pulpitis or photosynthetic trauma.
How did the authors differentiate between postoperative pain caused by pulpitis and pain resulting from occlusal interference or mechanical trauma?
Was cavity size standardized across specimens? In Figure 4, the cavity dimensions appear to differ between samples. Prior research has shown that cavity size can influence the pulp healing response following capping procedures.
(e.g., doi: 10.1111/iej.13099)
Inflammation is discussed based on HE staining in Figure 5. While we understand the authors' intent, HE staining alone is not a reliable method for evaluating inflammation in a scientific context. The use of additional techniques, such as immunohistochemistry, is recommended.
In the Discussion, the authors state that rat molars are approximately twice the size of mouse molars. However, based on our experience, the difference is closer to threefold. Please verify and revise accordingly.
The authors conclude: “Therefore, for future research, the use of this model is recommended in mature rats weighing between 350 and 400 grams.” We do not support this recommendation, as most widely accepted pulp exposure models employ younger rats.
Justify this deviation or revise the recommendation.
doi: 10.1016/j.joen.2023.11.009
doi: 10.1111/iej.13888.
doi: 10.1016/j.joen.2025.06.002.
How many evaluators assessed the behavioral responses? Were any inter-rater discrepancies observed, and if so, how were they addressed? Please include information on the inter-rater reliability of behavioral scoring.
The references provided are intended as a starting point for further discussion and are not mandatory for citation. Please refer to them if you believe they support your revisions, or feel free to cite alternative, appropriate sources that better align with your manuscript.
Author Response
Reviewer 1:
We sincerely thank you for your valuable comments and observations, which have significantly contributed to strengthening and enriching our manuscript. Below, we provide detailed responses to each of your questions and suggestions. In the revised version of the article, the changes made in response to your feedback are highlighted in yellow, and we indicate the specific sections and respective lines where these modifications can be found, so they can be easily identified.
This manuscript presents a postoperative behavioral evaluation using a mechanically induced pulpitis model. While the study falls within the scope of Biomedicine, there are several concerns that should be addressed to improve the scientific rigor and clarity of the manuscript.
- Regarding the materials chosen, amalgam and zinc phosphate cement are generally not considered biocompatible with the dental pulp. They are not treated as direct pulp cappping materials.
The authors should clearly justify their use in this study.
We do not use amalgam and zinc phosphate cement as direct pulp capping materials, precisely because, as you rightly pointed out, they are not biocompatible with dental pulp tissue. Our purpose was not to protect the pulp but rather to induce a controlled, pathogen-free, and consistent inflammatory process. Our goal was achieved by ensuring pulp injury through both physical and chemical means: physically, by entering the cavity with a bur and using a file to confirm access to the pulp chamber; and chemically, through the sealing material, given that its chemical nature can modulate the intensity of the inflammatory response. Amalgam was selected due to its potential to induce mild and chronic inflammation through the release of metal ions, particularly when in direct contact with the pulp. In contrast, zinc phosphate cement is known to cause more severe inflammation, primarily due to its acidity. The premise described previously is supported by the following findings from the literature:
Chandwani et al. conducted a study on human premolars with orthodontic indications. Under anesthesia they prepared class II cavities and restored 50 teeth with amalgam and 50 with resin, evaluating the inflammatory response at 24 hours and 7 days. At each time point, the teeth were extracted and subjected to histological analysis. Among the 25 amalgam-restored teeth evaluated at 24 hours, 32% showed a mild pulp response, 48% a moderate response, and 20% a severe inflammatory response. At 7 days, in a separate group of 25 amalgam-restored teeth, 60% exhibited a mild response, while 40% showed a moderate to severe response. Fibrosis was observed in approximately 8 teeth (32%), abscess formation in 2 teeth, and necrosis in 3 teeth. Based on these findings, the authors concluded that the use of amalgam may lead to prolonged inflammation, posing a risk of tissue degradation, fibrous tissue formation, or necrosis, depending on the intensity and duration of the chemical stimulus. (doi: 10.1159/000355607).
In another human study aimed at investigating the effect of caries and its restoration using temporary (including zinc phosphate) and standard permanent (including amalgam) filling materials on a panel of 16 inflammatory (cytokine) and oxidative markers in the gingival crevicular fluid of periodontally healthy individuals, samples were taken at 7 days (7D) and 30 days (30D) after restoration. Intact teeth served as the control group. The study found that the presence of these markers in crevicular fluid could reflect processes occurring at the pulp level. Specifically, teeth restored with zinc phosphate showed no substantial changes in cytokine levels compared to baseline; however, when compared to the control group, there was an increase in IL-2 and IL-4 at 7D and IFN-g at 30D. Additionally, there was an increase in oxidative markers such as GSH (reduced glutathione) and t-SOD (total superoxide dismutase) compared to baseline, with t-SOD at 7D showing significant differences relative to the control group. In contrast, teeth restored with amalgam did not show significant changes in key cytokines at any time point. However, there was a marked increase in GSH and t-SOD at both 7D and 30D, with t-SOD showing significant differences compared to the control group at both time points. In the discussion, the authors highlight the observed effects of zinc phosphate, suggesting it may stimulate odontoblast activity and increase the release of proinflammatory mediators. Furthermore, increased dentinal tubule permeability could facilitate acid perfusion, triggering a reactive inflammatory response in odontoblasts and dental pulp. On the other hand, mercury (Hg2+) and nickel (Ni2+) ions may induce oxidative stress and chronic inflammation even at non-cytotoxic concentrations, potentially leading to clinical signs at the pulp level (doi: 10.3389/fimmu.2021.716359).
Previous research has reported that the acidic (low pH) components of cements can cause pulp pain and inflammation, potentially leading to necrosis (PMID: 8210321). Zinc phosphate cement has been primarily used for crown placement and as an intermediate base (DOI: 10.1111/j.1834-7819.2010.01297.x). Researchers conducted a study in monkeys over a period of 3 to 21 days to determine whether the acidic components of zinc phosphate directly caused pulp inflammation or necrosis, or if the presence of bacteria was the critical factor. In this study, mechanically exposed pulp tissue was sealed directly with zinc phosphate, along with an additional biological seal using zinc oxide-eugenol. By day 10, the exposed pulps had healed and formed dentin bridges. The reorganized pulp tissue beneath the zinc phosphate cement exhibited mild inflammation, which was characterized over time according to the study’s evaluation criteria. At 5 days, a Grade 2 (mild) and Grade 3 (severe) inflammatory cell response was observed, along with Grade 2 (incomplete) soft tissue organization. Grade 3 corresponded to necrosis. By days 14 and 21, a Grade 1 response was observed for both inflammatory cells and tissue organization, indicating a return to normal tissue parameters. Based on the results of this experiment, where inflammatory responses were observed in sealed cavities as early as 5 days, it is possible to confirm the inflammatory states observed in our study. These were likely induced by the chemical components of the zinc phosphate filling material. Furthermore, the continued presence of the sealing material during the evaluation period likely limited microbial migration to the pulp tissue.
We did not identify any rodent animal models that used both zinc phosphate and amalgam as adjuvants of pulp injury. Therefore, we believe this study is the first to propose the use of traditionally employed dental materials that can induce non-pathological inflammatory effects due to their chemical composition in a rat model. Throughout the experimental evaluation period, the sealing material remained in place in the treated teeth.
These concepts have been incorporated into the latest version of the article, along with their corresponding scientific references: section introduction in lines 95 to 100; section discussion in lines: 684-696 and 707-712.
- The authors applied capsaicin to the exposed pulp, but the manuscript lacks essential details such as the concentration used, the volume applied, the physicochemical properties of the solution, and the specific application method.
Add detail information for reproducibility.
Before starting our study, the IACUC-PUJ requested and approved the performance of a pilot study to evaluate the presence of pain in the animals. Ten rats were used, each subjected to a different condition, and divided into two groups: one with lidocaine application (n=5) and one without lidocaine (n=5). Lidocaine was selected because, according to the literature, it is an anti-inflammatory agent that does not interfere with the modulation of inflammatory mediators relevant to the local immune response.
Each group included five individually tested conditions:
*Mechanically induced pulpitis with amalgam filling (MIPA)
*Mechanically induced pulpitis with zinc phosphate filling (MIPZ)
*Amalgam mechanically induced pulpitis + 0.5 ml CP 100 mM (MIPA+CP)
*Mechanically induced pulpitis with zinc phosphate + 0.5 ml CP 100 mM (MIPZ+CP)
*Negative control: rat only with lidocaine for the first group and rat without any experiment for the second group (NC).
When Capsaicin (CP) was used, the solution was prepared one day prior to the experiment under sterile conditions in a laminar flow cabinet. Starting with a stock solution of CP in ethanol at 327.400 mM, 3.05 mL was diluted to 10 mL with sterile-filtered PBS (pH=7.4) and stored at 4°C until use.
On the day of the procedure, the CP solution was transported to the operating room. Using a pipettor with a sterile tip, exactly 0.5 ml was applied to the exposed pulp. Immediately after perfusion, the cavity was sealed with either amalgam or zinc phosphate cement.
The operating room was environmentally controlled, with positive pressure and more than 25 air exchanges per hour. Incoming air passed through high-efficiency filters, and temperature was maintained between 22 and 26 °C and relative humidity 30 and 60%. The walls are smooth, with curved connections to the floor and ceiling, and the room was thoroughly disinfected before and after each procedure.
The operator verified the ingress and diffusion of the CP solution into the exposed pulp using an LED headlight with a 3.5X magnifying telescope (Dr. Kim Headlight, Innovaderma).
At the end of the pilot study, results showed that none of the ten rats exhibited signs of compromised
well-being. Therefore, it was justified to proceed with the main study without the use of lidocaine for pain management. However, the IACUC-PUJ requested that the study continue only with the mechanical induction of pulpitis to validate the model before introducing molecules with therapeutic potential such as capsaicin. The results presented in this publication correspond to the phase of the study.
The requested data were included in the lines 136 to 141 within the Materials and Methods section
- The cavity on mandibular molar was prepared mechanically and then exposed pulp was capped with materials. How was saliva contamination controlled during the procedure?
The authors should describe the measures taken to minimize contamination.
Previous studies have emphasized the importance of strict contamination control.
(e.g., doi: 10.1007/s00784-018-2374-5).
In the present study, the importance of preventing contamination of the surgical area with saliva during mechanical induction of pulpitis was emphasized through the implementation of the following measures:
- Head fixation: A custom-designed platform developed by the researchers, was used to prevent head movement (Figure 3B).
- Oral opening fixation: Elastic bands and orthodontic ligatures were anchored to posts on the platform to maintain the mouth open (Figure 3 G-L).
- Tongue retraction: A saliva ejector was used to isolate the tongue from the lower first molar.
- Facial muscle retraction: A dental spatula was used to retract the facial muscles (Figure 3F).
- Disinfection of the peridental area: A 0.2% chlorhexidine solution was applied using disposable Micro Applicators (Global Roll™).
- Tooth access: A high-speed handpiece activated by nitrogen was used to open the tooth, with continuous irrigation using sterile 0.9% (w/v) NaCl solution.
- Drying of the peridental area: Sterile cotton swabs were used to maintain dryness and isolate the area from saliva during the procedure.
Following steps a to g ensured that the surgical field remained free of saliva. Additionally, as previously mentioned, all materials used were sterile, and the operating room conditions supported sterility maintenance.
These aspects were incorporated into the Discussion section (Lines: 673 - 681), and the reference you recommended have been added and into Materials and Methods section MIP description (Lines 179-198) was refined with these aspects.
- After restoring the cavity with amalgam or zinc phosphate cement, how was occlusal adjustment verified?
Describe the method used to confirm that occlusal trauma was not introduced during the procedure. I am concerned that it is not possible to determine whether postoperative pain and behavioral changes are due to pulpitis or photosynthetic trauma.
How did the authors differentiate between postoperative pain caused by pulpitis and pain resulting from occlusal interference or mechanical trauma?
Occlusal trauma was not conventionally confirmed. However, during cavity preparation and filling, the dentist-endodontist ensured that the sealing material was adequately condensed and sealed inside the cavity, aiming to preserve the occlusal table and maintain the characteristic morphology of the first molar.
Since the animal was intubated under anesthesia, it was not possible to verify occlusion. However, this type of anesthesia provided the advantage of having sufficient time to perform the obturation optimally.
Post-operative well-being was assessed by observing the rats after the procedure. As in most cases, the animals had access to food and water immediately after surgical recovery, and no signs of discomfort were observed during feeding. Therefore, it was concluded that there was no postoperative pain. Consequently, occlusal or mechanical trauma or pulpitis could be ruled out (doi: 10.1016/j.joen.2009.09.029).
These aspects were incorporated into Materials and Methods section in Lines 195-198 and into Discussion section in Lines 698-703.
- Was cavity size standardized across specimens? In Figure 4, the cavity dimensions appear to differ between samples. Prior research has shown that cavity size can influence the pulp healing response following capping procedures.
(e.g., doi: 10.1111/iej.13099)
The size of the cavity was standardized and consistently prepared in the same manner from the beginning. All cavities were performed by the same calibrated endodontist operator to ensure consistency across specimens. In each subject, cavities were created using a sized ¼ round carbide bur in the mesial fossa of the lower right first molar, progressively advancing through the dentin until reaching the pulp chamber. Therefore, no substantial differences in cavity width or depth were observed among the specimens.
Regarding the comment, what is seen in Figure 4 is that Figure 4F appears larger, and the sealing material (amalgam) looks more radiopaque due to image distortion during the editing process, giving the impression that the cavity appears deeper and wider. The X-ray images directly correspond to the clinical and occlusal photographs, positioned above the radiographs. The purpose of presenting these paired images was to demonstrate that the sealing material remained intact throughout the experimental period, from MIP to euthanasia, not to illustrate cavity size. This was previously explained in the lines: 330 and 331 in the manuscript.
These aspects were incorporated into Materials and Methods section in Lines 183-187 and into Discussion section in Lines 679-681. The recommended reference was incorporated into the manuscript.
- Inflammation is discussed based on HE staining in Figure 5. While we understand the authors' intent, HE staining alone is not a reliable method for evaluating inflammation in a scientific context. The use of additional techniques, such as immunohistochemistry, is recommended.
Our IACUC required us to confirm the pulp injury through hematoxylin and eosin (H&E) staining. We performed this analysis; however, we also considered identifying different inflammatory cell types using confocal microscopy and immunohistochemistry to complement the findings related to inflammation during our experiments. Unfortunately, during the antigen retrieval procedures, the mandibular and dental samples detached, despite multiple evaluations and modifications to the tissue fixation technique on the slides, including the use of control tissues such as spleen, intestine, and tonsil.
These modifications included testing various types of slides (varying brands, coating materials, and manufacturing processes), as well as reviewing potential causes such as insufficient fixation, improper handling, or inadequate decalcification. Ultimately, we identified that the high temperature (90°C) used during antigen retrieval was the most likely cause of tissue detachment.
Our team includes experienced professionals, among them a pathologist and technicians who routinely process samples in the pathology department. Together, we concluded that confocal microscopy was not viable for this analysis, as pulp tissue is too small and delicate, preventing proper fixation or adhesion to the slides, unlike other tissues that respond well to standard fixation techniques (e.g., spleen, intestine, and tonsil).
Inflammation in our study was determined in two ways: first, by clinically observing coronal bleeding immediately after pulp exposure with the bur and upon insertion of the No. 15 file into the pulp chamber (figures 3 F, G, and H); and second, through H&E staining, which revealed discontinuity in the pulp horns due to injury and the presence of inflammatory cell populations, likely including neutrophils (Figures 5 E and F).
To strengthen these findings, we included Table 1 in the manuscript, describing the inflammatory status after MIPA and MIPZ using two parameters: the percentage of the coronal area affected by inflammatory infiltrate and the percentage of necrosis. Descriptive statistics (relative frequencies) were applied for each treatment at both evaluation time points.
With this complement, we consider hematoxylin and eosin staining to be a fundamental histological technique and a scientifically supported method for approximating evidence of an inflammatory process, as several direct precedents in the literature have employed it. As examples, we cite the following authors:
doi: 10.1016/j.joen.2016.09.003
doi: 10.1016/j.archoralbio.2017.08.002
doi: 10.1371/journal.pone.0207411
doi: 10.3390/biomedicines9070784
doi: 10.1177/0022034512454297
The described adjustments have been included in the abstract section (lines: 30-31, 33 and 34), materials and methods section (lines: 247 to 255), statistical analysis subsection (lines: 257 to 262); results section (lines 347-348, 375-394) and discussion section: (lines: 707 to 712).
- In the Discussion, the authors state that rat molars are approximately twice the size of mouse molars. However, based on our experience, the difference is closer to threefold. Please verify and revise accordingly.
Since our research group has experience only with rats, we searched the scientific literature for differences in molar size between rats and mice. The only reference found that provided a quantitative comparison was Christensen et al., 2023 (DOI: 10.1073/pnas.2300374120), who stated: “First lower molars of the mouse and the rat are similar in overall shape, but the rat molar is two times larger in linear dimensions. This reflects the body size difference between the species”.
This was documented in the discussion section, lines: 641 to 644.
- The authors conclude: “Therefore, for future research, the use of this model is recommended in mature rats weighing between 350 and 400 grams.” We do not support this recommendation, as most widely accepted pulp exposure models employ younger rats.
Justify this deviation or revise the recommendation.
doi: 10.1016/j.joen.2023.11.009
doi: 10.1111/iej.13888.
doi: 10.1016/j.joen.2025.06.002.
The purpose of using the rat model, from a translational medicine perspective, was to better understand inflammatory processes in humans, specifically pulpitis. In this study, our model focuses on inflammation rather than repair, as opposed to models using young rats, which are more commonly associated with regenerative responses.
We justified the use of adult rats by considering that exhibits a more prominent inflammatory response, which may resemble that of adult humans. The next step in our research is to propose potential immunomodulatory molecules capable of preserving tissue during inflammatory processes in adults.
Additionally, adult rats allow for unrestricted access to the dental instruments required to induce pulp injury, as well as for intubation, without compromising animal well-being during the procedure.
From lines 748 to 755 in the discussion section, the rationale of our research group was better structured, replacing “recommendation with “advantage” and “maturity” with “adult”.
The recommended reference was incorporated into the manuscript.
- How many evaluators assessed the behavioral responses? Were any inter-rater discrepancies observed, and if so, how were they addressed? Please include information on the inter-rater reliability of behavioral scoring.
In the present study, a pilot phase was first conducted in which pain-related behavior in rats with induced pulpitis was evaluated using the Rifai, Grimace and Flecknell scales.
All behavioral observations were independently performed by two trained evaluators, both blinded to treatment allocation. Prior to data collection, the evaluators participated in a calibration session using a subset of recordings to standardize scoring criteria.
Discrepancies, which accounted for less than 5% of the observations, were resolved through joint review and consensus based on video recordings. No significant discrepancies were observed between evaluators, and a kappa coefficient greater than 0.80 was obtained for all behaviors assessed.
During this phase, the IACUC required a thorough and independent review of the recordings by multiple observers in all cases where a score of zero was assigned. These results were reviewed and validated by our IACUC.
Based on these findings, our IACUC authorized the continuation of the study, allowing for the evaluation of rat behavior within the cage using forms incorporating the aforementioned scales (line 213 and 604-605). Although the scoring system allows for the classification of behaviors as mild, moderate, or substantially severe, no pain-related behaviors (e.g. head shaking, facial grimacing) were identified upon consensus review, confirming the original score of “0” for all pain parameters. It is important to note that the pain scales and scoring system used in this study are discrete, which may limit the detection of subtle changes in pain behavior.
The described adjustments have been included in materials and methods section (lines: 214 to 222); and discussion section: (lines: 610 to 614).
- The references provided are intended as a starting point for further discussion and are not mandatory for citation. Please refer to them if you believe they support your revisions, or feel free to cite alternative, appropriate sources that better align with your manuscript.
We appreciate the references you shared; we reviewed each one and incorporated those we considered aligned with our manuscript.
Reviewer 2 Report
Comments and Suggestions for Authors
For Abstract:
The abstract should summarize all parts of the experiment, including histological evaluation and primary outcomes (e.g., inflammation reports, etc.).
In the study on rats, they were underwent mechanical induction of pulpitis (MIP) under anesthesia with and without lidocaine. is it ok that they prescribed lidocaine systemically during the procedure at a dose of 0.67-121 1.3mg/Kg/h, and again one-hour post-procedure at 0.17 mg/kg over 3 minutes for maintenance and control of pain? or visit more common to administer locally?
- method
Line 166: Whether the same ¼ round bur created the cavity and exposure, or if secondary tools were used. How was exposure size standardized?
lines 121-122: The use of systemic lidocaine as used in your pilot study, although feasible, does not follow standard local anesthetic protocols used in the management of clinical pulpitis. Please explain why this route was chosen rather than a local route (e.g., nerve block). Furthermore, given the systemic anti-inflammatory properties of lidocaine, how was its potential impact on neuroimmune responses controlled? References to previous studies using similar regimens of systemic lidocaine in similar models would strengthen the methodological justification.
A compound with therapeutic potential Capsaicin (CP), was applied prior to sealing the cavity with each material, resulting in the subgroups X+CP and Y+CP. But it is not clear the intension and the finding of these subgroups that they introduced.
Line 126: The authors introduced capsaicin (CP) as pretreatment before cavity sealing (subgroups MIPA+CP and MIPZ+CP), but did not clearly state the scientific rationale for this intervention. What was the hypothesized mechanism of CP in this model? Was it supposed to modulate pain, inflammation, or bacterial effects? Furthermore, it is unclear whether CP was administered in the original study.
Provide the number of rats that underwent euthanasia, radiography, and histological evaluation. If only a few specimens were used, state the number.
Results:
There are two key concerns about the statistical analysis: (1) the use of the Kruskal-Wallis test with a severely unbalanced design (n=10 vs. n=2) carries the risk of unreliable conclusions due to insufficient power and rank distortion in the small control group. (2) I am not sure that the Bartlett test for homogeneity of variances is appropriate here. Given these limitations, the authors should consult a statistician to evaluate their approach.
line 392: Please provide informative caption for table 4.
probably “day before MIP, the day of the procedure, and days 1, 2 and 5 post procedure” correspond to i, 1,2,3, and 6?
Discussion:
Line 477: The discussion should explicitly compare the advantages and differences of your mechanical pulp injury model with the models described by Li et al. (2025).
Lines 508-509: The statement that all pain-related parameters (facial expression and abnormal head or body movements) received a score of "0" contradicts established pulpitis pain models and the statement in your line 207. If animals showed pain behaviors but were incorrectly scored as "0", correct the results and address the implications.
Author Response
Reviewer 3:
We sincerely thank you for your valuable comments and observations, which have significantly contributed to strengthening and enriching our manuscript. Below, we provide detailed responses to each of your questions and suggestions. In the revised version of the article, the changes made in response to your feedback are highlighted in yellow, and we indicate the specific sections and respective lines where these modifications can be found, so they can be easily identified.
- The abstract should summarize all parts of the experiment, including histological evaluation and primary outcomes (e.g., inflammation reports, etc.).
Histological evaluation and primary outcomes (e.g., inflammation reports, etc.) were included in the abstract, lines 30-31, 33-34 and further detailed in the Materials and Methods section in lines: 247- 255 and 257-262; the Results section: lines 347-348 and 375-394, including table 1, and the Discussion section: lines 704-709.
- In the study on rats, they were underwent mechanical induction of pulpitis (MIP) under anesthesia with and without lidocaine. is it ok that they prescribed lidocaine systemically during the procedure at a dose of 0.67-121 1.3mg/Kg/h, and again one-hour post-procedure at 0.17 mg/kg over 3 minutes for maintenance and control of pain? or visit more common to administer locally?
Before starting our study, the IACIC-PUJ requested and approved the implementation of a pilot to evaluate the presence of pain in the animals. Ten rats were used, each subjected to a different condition, and divided into two groups: one with lidocaine application (n=5) and one without lidocaine (n=5).
Lidocaine was selected because, according to the literature, due to its anesthetic effect at low doses, both in systemic – subcutaneous (SC) and embolic applications, which provides controlled analgesia in animal models (DOI: 10.1016/S0304-3959(02)00028-3) (DOI: 10.1111/j.1467-2995.2009.00480.x). Since evaluating the pulpal inflammatory response, specifically the presence of cells and molecules such as cytokines and neuropeptides, was crucial for us, the systemic control at low doses was seen as an advantage, as it would not have a direct or substantial effect on modifying the local response. Currently, there is no scientific evidence support in the use of systemic lidocaine (SC) to control pulpal pain in rat models.
During the development of the pilot study and subsequent analysis of the documented results, the IACUC approved that the use of lidocaine was not necessary, as the group of animals without lidocaine did not show any signs of compromised welfare. Therefore, we were authorized to proceed without lidocaine, and consequently, the main study did not use lidocaine.
The described adjustments have been included in Materials and Methods section (lines: 132) results section: (lines: 314-315).
- Method: Line 166: Whether the same ¼ round bur created the cavity and exposure, or if secondary tools were used. How was exposure size standardized?
The size of the cavity was standardized and consistently prepared in the same manner from the beginning. All cavities were performed by the same calibrated endodontist operator to ensure consistency across specimens. In each subject, cavities were created using a sized ¼ round carbide bur in the mesial fossa of the lower right first molar, progressively advancing through the dentin until reaching the pulp chamber. Therefore, no substantial differences in cavity width or depth were observed among the specimens.
These aspects were incorporated into Materials and Methods section in Lines 183-187 and into Discussion section in Lines 679-681.
- Method: lines 121-122: The use of systemic lidocaine as used in your pilot study, although feasible, does not follow standard local anesthetic protocols used in the management of clinical pulpitis. Please explain why this route was chosen rather than a local route (e.g., nerve block). Furthermore, given the systemic anti-inflammatory properties of lidocaine, how was its potential impact on neuroimmune responses controlled? References to previous studies using similar regimens of systemic lidocaine in similar models would strengthen the methodological justification.
Many of these questions were also addressed in point 2.
It is important to understand that systemic anesthesia is mandatory to ensure proper control of the animal and to carry out all oral procedures. It provides the operator with sufficient time to perform the experimental tasks correctly and improves the animal’s well-being; this is a requirement of the IACUC. Additionally, the use of lidocaine was reinforced during the postoperative period due to its analgesic effect, provided it is administered at low doses.
We did not administer local (inferior alveolar nerve block) anesthesia to avoid altering the pulpal inflammatory response induced by MIP, the main purpose of this work.
During post-MIP observation, some animals showed mild numbness that did not last more than one hour due to the anesthetic effect. After that period, they behaved normally and continued to take food and water.
Refinement aims to reduce pain and distress in animals so they can maintain their well-being. Among the strategies it includes are staff training and skill development for animal handling. Our personnel have been trained to administer substances via subcutaneous and intravenous routes in rats, which ensures the systemic anesthetic effect of lidocaine and replaces the need for local anesthesia.
- A compound with therapeutic potential Capsaicin (CP), was applied prior to sealing the cavity with each material, resulting in the subgroups X+CP and Y+CP. But it is not clear the intension and the finding of these subgroups that they introduced.
- Line 126: The authors introduced capsaicin (CP) as pretreatment before cavity sealing (subgroups MIPA+CP and MIPZ+CP) but did not clearly state the scientific rationale for this intervention. What was the hypothesized mechanism of CP in this model? Was it supposed to modulate pain, inflammation, or bacterial effects? Furthermore, it is unclear whether CP was administered in the original study.
As mentioned in point 2, our IACUC requested and approved the use of a pilot study to evaluate animal well-being in relation to various potentially affecting variables. Among these, we assessed whether the model would allow the addition of a volume of a molecule with therapeutic potential (e.g., CP), without altering the stability of the sealing material during its condensation and final hardening process resulting from its chemical reaction and ensuring that the material remained stable and fixed within the cavity throughout the duration of the experiment. In this case, capsaicin was tested, and as in all other cases, no loss of well-being was observed. It was not used in the main study.
- Provide the number of rats that underwent euthanasia, radiography, and histological evaluation. If only a few specimens were used, state the number.
Number of rats included:
Pilot study: 10 rats
Main study: 44 rats
Euthanasia: 54 rats
X-rays: 25 radiographs
Histology: 54 samples
- Results: There are two key concerns about the statistical analysis: (1) the use of the Kruskal-Wallis test with a severely unbalanced design (n=10 vs. n=2) carries the risk of unreliable conclusions due to insufficient power and rank distortion in the small control group. (2) I am not sure that the Bartlett test for homogeneity of variances is appropriate here. Given these limitations, the authors should consult a statistician to evaluate their approach.
We completely agree with you, and we respond precisely with the adjustments we have made:
The negative control group (NC; n=2) has been excluded from all inferential statistical tests and will be used solely for descriptive purposes. The highly unbalanced design was intentionally implemented in accordance with the 3R principle (Replace, Reduce, Refine) for animal research, minimizing the number of animals used in the control group while still providing sufficient descriptive validation of baseline behavior. Therefore, the inferential comparisons of interest were conducted exclusively between the experimental MIPA and MIPZ groups (n=10 vs. n=10), which were determined a priori to achieve 81% power to detect a large effect size (d=0.8). By limiting the analysis to a balanced design (10 vs. 10), we avoid the low power and rank distortion that could result from including a group of n=2 in non-parametric tests.
To strengthen the robustness of our approach, we propose adding the following to the Methods section:
* Complementing Bartlett’s test with Levene’s (or Brown–Forsythe) test to assess variance homogeneity under conditions where the normality assumption is not met.
* Retaining Bartlett’s test for normally distributed data and stipulating that, in case of discrepancies, Levene’s test results will take precedence.
We recognize that, when assigning ranks across all data, a very small group (e.g., NC; n=2) can disproportionately occupy extreme ranks, biasing the H statistic.
In our analysis, we excluded the negative control group (n=2) from inferential tests and compared only MIPA and MIPZ (n=10 vs. n=10), ensuring uniform rank distribution and eliminating rank distortion.
Therefore, we have made these modifications to ensure the validity and reliability of our result:
* For each outcome (body weight, food consumption, water consumption, and their changes at each time point), we now apply Levene’s test to verify homogeneity of variances between the two treatment groups (MIPZ vs. MIPA).
In general, the following methodological algorithm was applied:
If both (a) Shapiro–Wilk tests confirm normality in each group and (b) Levene’s test indicates equal variances (p > 0.05), we perform a two-sample Student’s t-test. - If normality holds but variances differ (Levene’s p ≤ 0.05), we use Welch’s t-test. And if normality is violated in either group (p ≤ 0.05), we employ the Wilcoxon rank-sum test (Mann–Whitney U).
We believe this streamlined approach addresses concerns about power, rank distortion, and appropriateness of tests given our balanced inferential design (n = 10 vs. 10).
These aspects were incorporated into Materials and Methods section, Statistical analysis in Lines 264-274 and into Results section in Lines: 406-415, 431-437, 444, 460-463,
- Results: line 392: Please provide informative caption for table 4.
probably “day before MIP, the day of the procedure, and days 1, 2 and 5 post procedure” correspond to i, 1,2,3, and 6?
Thank you. This information has been included in lines: 466-479.
- Discussion: Line 477: The discussion should explicitly compare the advantages and differences of your mechanical pulp injury model with the models described by Li et al. (2025).
Animal models are essential for exploring the underlying mechanisms of oral infectious diseases and for evaluating potential therapeutic strategies. Several models have been developed to induce pulpitis, including open mechanical pulp exposure, lipopolysaccharide (LPS) stimulation, caries-induced models, laser-induced models, and transgenic mouse models.
It is crucial to employ models that ensure animal well-being by closely monitoring changes in body weight, food and water intake, and behavioral activity. Among these, the caries, induced model requires a carbohydrate-rich diet, which over time can lead to obesity and diabetes. Additionally, it involves the inoculation of one or more bacterial strains to promote a diverse microbial environment and often necessitates surgical recession of salivary glands to reduce salivary flow and facilitate caries development.
The transgenic pulpitis model involves the conditional overexpression of TNF-a in dental pulp and bone, leading to pulp inflammation that mimics pulpitis. However, this model may limit the full expression of inflammatory mediators, potentially restricting its translational relevance.
In contrast, the mechanical pulp exposure model, when sealed with materials such as amalgam or zinc phosphate cement, induces a chemical challenge without significantly compromising animal welfare, this is crucial to ensure valid outcomes in studies of inflammation, pain or immunology. This model offers a valuable approach for studying the cellular and molecular mechanisms of pulp inflammation and for evaluating the immunomodulatory potential of candidate therapeutic agents.
The described adjustments have been included in discussion section: (lines: 562 to 573 and 584-585).
- Lines 508-509: The statement that all pain-related parameters (facial expression and abnormal head or body movements) received a score of "0" contradicts established pulpitis pain models and the statement in your line 207. If animals showed pain behaviors but were incorrectly scored as "0", correct the results and address the implications.
In the present study, a pilot phase was first conducted in which pain-related behavior in rats with induced pulpitis was evaluated using the Rifai, Grimace and Flecknell scales.
All behavioral observations were independently performed by two trained evaluators, both blinded to treatment allocation. Prior to data collection, the evaluators participated in a calibration session using a subset of recordings to standardize scoring criteria.
Discrepancies, which accounted for less than 5% of the observations, were resolved through joint review and consensus based on video recordings. No significant discrepancies were observed between evaluators, and a kappa coefficient greater than 0.80 was obtained for all behaviors assessed.
During this phase, the IACUC required a thorough and independent review of the recordings by multiple observers in all cases where a score of zero was assigned. These results were reviewed and validated by our IACUC.
Based on these findings, our IACUC authorized the continuation of the study, allowing for the evaluation of rat behavior within the cage using forms incorporating the aforementioned scales (line 209 and 585). Although the scoring system allows for the classification of behaviors as mild, moderate, or substantially severe, no pain-related behaviors (e.g. head shaking, facial grimacing) were identified upon consensus review, confirming the original score of “0” for all pain parameters. It is important to note that the pain scales and scoring system used in this study are discrete, which may limit the detection of subtle changes in pain behavior.
The described adjustments have been included in materials and methods section (lines: 214 to 222); and discussion section: (lines: 610 to 614).
Reviewer 3 Report
Comments and Suggestions for Authors
1. Abbreviation Formatting
Please ensure consistency in abbreviation usage throughout the manuscript. For example, "SPF" should be written as "Specific pathogen-free (SPF)" following standard terminology. Additionally, "SpOâ‚‚" should be accurately defined as "Peripheral capillary oxygen saturation (SpOâ‚‚)" instead of the more general term "Oxygen Saturation."
2. Sex of Animals
The manuscript does not mention the reason for excluding female rats. Given the potential influence of sex-specific biological differences on inflammation and pain response, please justify the exclusive use of male rats or acknowledge it as a limitation.
3. Missing Scale Bars
Several figure panels appear to lack a visible scale bar, which is essential for interpretation of microscopic images. Please revise all figures to include scale bars and ensure they are clearly labeled in the figure legends.
4. Group Assignment and Body Weight Variability
While the authors state that rats were "randomly assigned" into experimental groups, the baseline body weight data suggest considerable differences among groups. Specifically, the NC group had a mean body weight of 316 g (SD 7.36), whereas the MIP-A and MIP-ZP groups had higher and more variable weights: 382.5 g (SD 33.56) and 394.1 g (SD 32.8), respectively.
This discrepancy raises concerns about the adequacy of randomization and whether body weight was matched across groups. Given that body weight may influence pharmacokinetics or inflammation-related outcomes, the authors should clarify the randomization method and whether any stratification by weight was performed. If not, this should be acknowledged as a limitation.
Author Response
Reviewer 2:
We sincerely thank you for your valuable comments and observations, which have significantly contributed to strengthening and enriching our manuscript. Below, we provide detailed responses to each of your questions and suggestions. In the revised version of the article, the changes made in response to your feedback are highlighted in yellow, and we indicate the specific sections and respective lines where these modifications can be found, so they can be easily identified.
- Abbreviation Formatting
Please ensure consistency in abbreviation usage throughout the manuscript. For example, "SPF" should be written as "Specific pathogen-free (SPF)" following standard terminology. Additionally, "SpOâ‚‚" should be accurately defined as "Peripheral capillary oxygen saturation (SpOâ‚‚)" instead of the more general term "Oxygen Saturation."
Abbreviations were reviewed throughout the manuscript, as well as in the abbreviations table at the end, and consistency was maintained across all instances. Lines: 31, 111 to 112, 167 to 168, 325 to 326, 331.
- Sex of Animals
The manuscript does not mention the reason for excluding female rats. Given the potential influence of sex-specific biological differences on inflammation and pain response, please justify the exclusive use of male rats or acknowledge it as a limitation.
Males were selected to achieve greater homogeneity among the individuals assessed in the well-being study. Food consumption differs between males and females primarily due to body size, with males tending to be larger. Including both sexes could have resulted in variability in daily food intake, which was a key variable in our study.
Additionally, based on observations of our Lewis rat colony, males exhibit calmer behavior when handled by operators, which is advantageous for behavioral assessments. Although few studies address this topic, some—such as Zorczeniewska (2017)—have shown that stress moderately increases pain-like behaviors in females compared to males (DOI: 10.11607/ofph.1807).
The described adjustments have been included in discussion section: (lines: 634 to 638)
- Missing Scale Bars
Several figure panels appear to lack a visible scale bar, which is essential for interpretation of microscopic images. Please revise all figures to include scale bars and ensure they are clearly labeled in the figure legends.
The histological images with the requested scale bars were included in each photograph, along with their corresponding description in the figure legens of Figure 5.
- Group Assignment and Body Weight Variability
While the authors state that rats were "randomly assigned" into experimental groups, the baseline body weight data suggest considerable differences among groups. Specifically, the NC group had a mean body weight of 316 g (SD 7.36), whereas the MIP-A and MIP-ZP groups had higher and more variable weights: 382.5 g (SD 33.56) and 394.1 g (SD 32.8), respectively.
This discrepancy raises concerns about the adequacy of randomization and whether body weight was matched across groups. Given that body weight may influence pharmacokinetics or inflammation-related outcomes, the authors should clarify the randomization method and whether any stratification by weight was performed. If not, this should be acknowledged as a limitation.
In relation to group assignment and baseline body weight, we clarified our randomization procedure, demonstrated that baseline weights did not differ between the two main groups, and we acknowledged the absence of weight stratified randomization as a limitation.
Bellow, we explain the randomization method used:
All male Lewis rats (350–400 g) were first enrolled based on the predefined inclusion weight range to limit variability.
Within that range, animals were randomly assigned to treatment arms using a computer‑generated random sequence, with assignment concealed until immediately before the procedure.
We trust these additions clarify the adequacy of our randomization and address concerns about baseline variability.
The described adjustments have been included in materials and methods section (Lines 150 to 154) in discussion section: (lines: 626 to 627).
Round 2
Reviewer 1 Report
Comments and Suggestions for Authors
Reviewer comments,
Most of my concerns have been addressed by the author.
However, I would like to point out to the author that I have confirmed using micro-CT that the mesiodistal diameter of the occlusal surfaces of mouse molars is 1 mm, while that of rats is 3 mm. However, since this is not the essence of the content of this manuscript, I will not mention it in this peer review round.
Reviewer 2 Report
Comments and Suggestions for Authors
I have reviewed the revised manuscript and find it significantly improved compared to the previous version. You have addressed all previous comments clearly and satisfactorily.